# Crosstalk between Regnase-1 and -3 shapes mast cell survival and cytokine expression

Marian Bataclan[1], Cristina Leoni[1], Simone G Moro[1], Matteo Pecoraro[1], Elaine H Wong[2], Vigo Heissmeyer[2,3], Silvia Monticelli[1]

Post-transcriptional regulation of immune-related transcripts by RNA-binding proteins (RBPs) impacts immune cell responses, including mast cell functionality. Despite their importance in immune regulation, the functional role of most RBPs remains to be understood. By manipulating the expression of specific RBPs in murine mast cells, coupled with mass spectrometry and transcriptomic analyses, we found that the Regnase family of proteins acts as a potent regulator of mast cell physiology. Specifically, Regnase-1 is required to maintain basic cell proliferation and survival, whereas both Regnase-1 and -3 cooperatively regulate the expression of inflammatory transcripts upon activation, with *Tnf* being a primary target in both human and mouse cells. Furthermore, Regnase-3 directly interacts with Regnase-1 in mast cells and is necessary to restrain Regnase-1 expression through the destabilization of its transcript. Overall, our study identifies protein interactors of endogenously expressed Regnase factors, characterizes the regulatory interplay between Regnase family members in mast cells, and establishes their role in the control of mast cell homeostasis and inflammatory responses.

## Introduction

Mast cells are tissue-resident immune cells that exert key effector functions during allergic and anaphylactic reactions, act as sentinels against invading pathogens, and regulate physiological processes, such as tissue repair and angiogenesis (Bischoff, 2007; Theoharides et al, 2015). These cells are activated by a variety of signals through multiple receptors, most commonly via the high-affinity IgE receptor, FcεRI. Recognition of antigens or allergens by IgE antibodies bound to FcεRI triggers cell activation, characterized by degranulation and rapid release of a broad panel of pre-stored and de novo–synthesized mediators (histamine, prostaglandin, cytokines, proteases, etc.), eventually resulting in an inflammatory response (Reber et al, 2015; Theoharides et al, 2015). Dysregulation of mast cell activation and functions may lead to excessive and damaging responses to otherwise harmless agents, as exhibited in mast cell–activating syndromes and IgE-associated disorders (food allergy, allergic rhinitis, asthma, and anaphylaxis) (Theoharides et al, 2015). Tight control of mast cell functionality is therefore crucial to facilitate responses that will effectively eradicate a pathogen or allergen, while minimizing harm to the host.

Post-transcriptional regulation of immune-related mRNAs is critical in controlling immune cell responses. This can be mediated by RNA-binding proteins (RBPs), which bind to specific elements within the RNA transcripts and regulate different post-transcriptional events, such as RNA processing, modification, stability, and translation (Kafasla et al, 2014; Turner & Díaz-Muñoz, 2018; Bataclan et al, 2021). Mechanistically, RBPs can restrain inflammation by binding to *cis*-regulatory regions, such as AU-rich regions or stem–loop structures, in the 3′ untranslated region (UTR) of inflammatory mRNAs, leading to the destabilization and eventual degradation of these transcripts. In mast cells, we recently showed that mRNA methylation is crucial to restrain mast cell responses and that m$^6$A methylation of the *Il13* transcript determines its stability and the extent of IL-13 cytokine production (Leoni et al, 2023). However, the mechanistic details regarding the functions of many RBPs and their role in the regulation of mast cell responses are yet to be elucidated.

Here, we found that the expression of members of the Regnase family of RBPs is highly induced upon acute stimulation of mast cells. Regnase proteins (Regnase-1 to Regnase-4, encoded by the *Zc3h12a-d* genes) are essential to restrain excessive inflammatory responses in immune cells, through the action of an intrinsic RNase enzymatic activity (Uehata et al, 2013; Mino et al, 2015). Upon recognition of stem–loop structures in target mRNAs, Regnases directly cleave and destabilize inflammatory transcripts (Uehata et al, 2013). Linked to such a prominent role in restraining effector immune responses, Regnase-1 has recently gained attention as a candidate therapeutic target to enhance anti-tumor responses of CD8[+] T lymphocytes. Deletion of Regnase-1 was sufficient to improve T-cell expansion, functionality, and persistence, leading to

[1]Institute for Research in Biomedicine, Università della Svizzera Italiana, Bellinzona, Switzerland [2]Institute for Immunology, Biomedical Center, Faculty of Medicine, Ludwig-Maximilians-Universität in Munich, Planegg-Martinsried, Germany [3]Research Unit Molecular Immune Regulation, Helmholtz Zentrum München, Munich, Germany

Correspondence: silvia.monticelli@irb.usi.ch

enhanced anti-tumor responses in mice (Wei et al, 2019; Zheng et al, 2021; Mai et al, 2023; Raj et al, 2023). On the contrary, significant amelioration of disease was observed in preclinical models of autoimmunity in which the expression of Regnase-1 was stabilized by the in vivo injection of morpholino oligonucleotides. These oligonucleotides effectively blocked the stem–loop structures in the Zc3h12a 3′UTR that are normally targeted by Regnase-1 itself in a process of autoregulation, resulting in the observed amelioration of the disease (Tse et al, 2022). Within the myeloid compartment, Regnase-1 was shown to prominently regulate a number of inflammatory transcripts in macrophages, including Il6, Il12b, and Ptgs2 (Matsushita et al, 2009; Iwasaki et al, 2011; Mino et al, 2015). Differently from Regnase-1, which is expressed both in lymphoid and in myeloid cells, Regnase-3 functions primarily in myeloid cells, and its macrophage-specific deletion led to excessive IFN-γ expression (von Gamm et al, 2019). Despite the growing interest in understanding the role of Regnases in immune response regulation, numerous questions persist regarding their mechanism of action. These include the extent of their unique or redundant functions in immune cells, and whether they physically interact to modulate shared mRNA targets.

By leveraging a variety of experimental approaches, such as gene editing and genetic deletions of Regnase proteins, modulating their expression via mRNA delivery, and identifying their protein interactome through immunoprecipitation coupled with mass spectrometry, our study reveals that Regnase-1 serves as a key negative regulator of mast cell proliferation and survival. In contrast, Regnase-3 primarily modulates the expression of Regnase-1 within mast cells, with which it also physically interacts. Despite showing some distinct regulatory roles, influenced in part by the preference of these two proteins for different subcellular compartments, both Regnase-1 and -3 share regulatory control over a subset of inflammatory transcripts, including Tnf. In summary, our study delineates the protein interactome of endogenously expressed Regnase proteins and elucidates their significance in governing mast cell homeostatic and inflammatory responses.

# Results

### The expression of both Regnase-1 and -3 is induced in acutely stimulated mast cells

To determine how the expression of RBP-encoding genes changes in mast cells upon stimulation with IgE and antigen complexes, we first re-analyzed RNA-seq datasets of murine bone marrow–derived mast cells (BMMCs) stimulated for 2 h with IgE and antigen complexes from Li et al (2021), focusing only on RBP-encoding genes as defined in the RBP2go database (Caudron-Herger et al, 2021) and including only RBPs detected in more than one dataset (Fig S1A and Table S1). This resulted in overall 827 differentially expressed transcripts, of which 435 were associated with general RNA metabolic processes at gene ontology (GO) analysis. Within this category, RBP transcripts divided into further functional categories linked primarily to RNA processing, translation, and stability (Fig S1B). Next, we found that Zc3h12c (Regnase-3) was among the most

highly induced RBP-encoding transcripts after stimulation (Fig S1A). Two other Regnase family members, Zc3h12a (Regnase-1) and Zc3h12d (Regnase-4), were also induced, whereas Zc3h12b (Regnase-2) was downmodulated. Re-analysis of ATAC-seq data from Li et al (2021) also showed increased accessibility around the Zc3h12a and Zc3h12c promoters upon IgE crosslinking, suggesting transcriptional activation after acute stimulation (Fig S1C). Analysis of the expression of Zc3h12a-d in mast cells extracted ex vivo from different mouse tissues (ImmGen database) (Dwyer et al, 2016) also revealed the high expression of Zc3h12c in most tissue-derived mast cells (notably from the skin, less prominently from the peritoneal cavity), followed by the modest expression of Zc3h12a and very low levels of Zc3h12b and Zc3h12d (Fig S1D). Finally, data mining of published RNA-seq data from human mast cells revealed induction of ZC3H12A and ZC3H12C expression in stimulated mast cells obtained from the human skin (Gao et al, 2023). Similar results were obtained by RNA-seq of stimulated human peripheral blood–derived mast cells (Cildir et al, 2019) (Fig S1E). Overall, these observations point toward a role of selected Regnase family members in modulating mast cell functions, and we therefore focused our attention on this family of RBPs.

To validate and extend these findings, we first measured the expression of the Regnase-encoding transcripts in BMMCs (~99% pure populations, determined by FcεRIα and c-Kit staining; Fig S1F) activated with IgE and antigen complexes for the indicated times. Zc3h12b and Zc3h12d were expressed at very low levels and were not significantly induced upon stimulation, whereas both Zc3h12a and Zc3h12c were expressed and highly induced at early time points (within the first 1–2 h) in activated mast cells (Fig 1A). Similar significant induction specifically of Zc3h12a and Zc3h12c was observed in mast cells separated ex vivo from the peritoneal cavity and stimulated with PMA and ionomycin (Figs 1B and S1G). Regnase-1 and -3 were also induced at the protein level upon activation of BMMCs and peritoneal cavity–derived, in vitro–expanded mast cells (Fig 1C and D), pointing toward a role of these two Regnase family members in regulating mast cell responses to stimuli. We also detected the expected proteolytic cleavage of Regnase-1 upon activation, consistent with Malt-1 activation downstream of the IgE receptor complex (Klemm et al, 2006; Uehata et al, 2013). Whether Regnase-3 is regulated post-translationally remains to be understood, especially considering the complex pattern revealed by Western blot (Fig 1C and D), as reported also in the literature (von Gamm et al, 2019). In terms of subcellular localization, both Regnase-1 and -3 were detected within punctuated structures in the cytoplasm of acutely activated mast cells, with a limited overlap (Fig S2). Because mast cells can be activated by a large variety of signals, we determined whether stimuli other than IgE and antigen complexes affected Regnase expression. We found that both Zc3h12a and Zc3h12c were induced, albeit at different extents, by inflammatory signals that induce cytokine expression in mast cells, including IL-33 and LPS, but not by the c-Kit ligand stem cell factor (SCF), which by itself did not induce cytokine expression (Figs 1E and S3). Overall, these data point toward a role of Regnase-1 and -3 in modulating mast cell activation and/or effector functions that we set out to investigate.

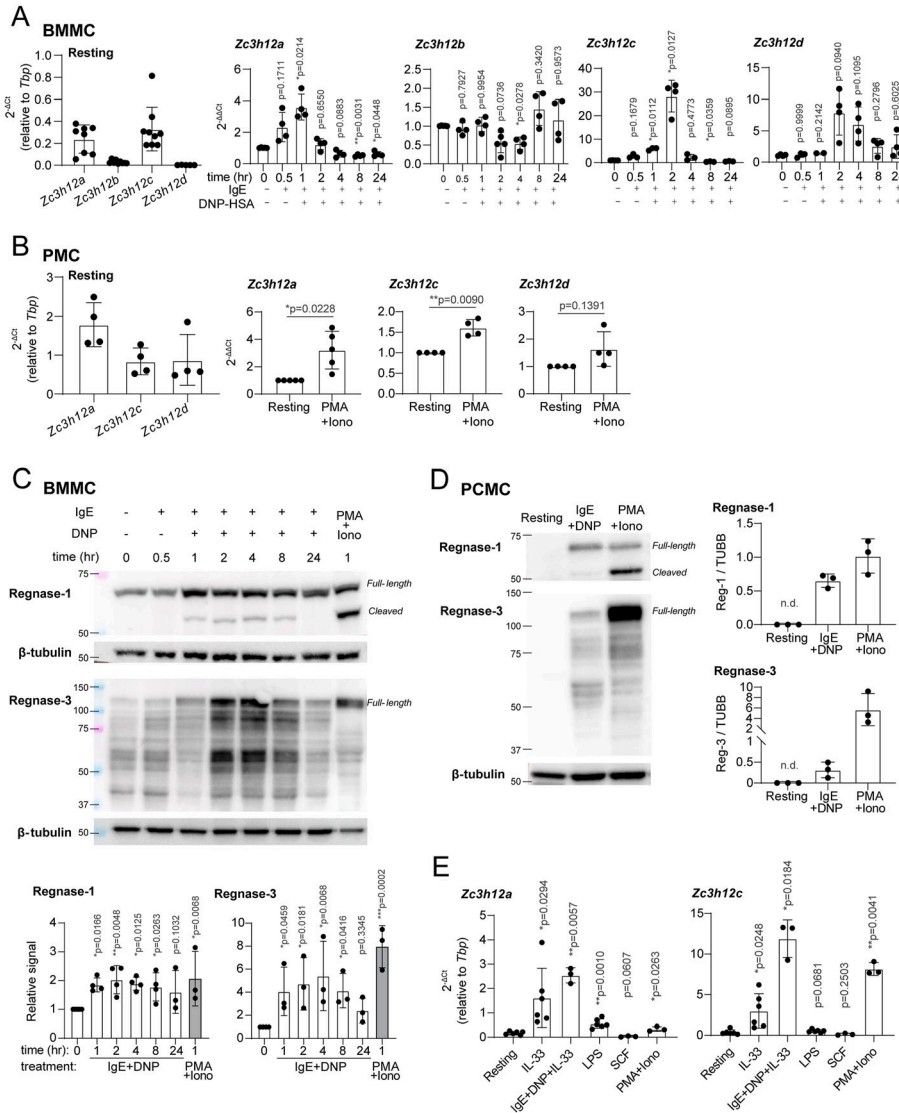

**Figure 1. Expression of Regnase-1 and Regnase-3 is induced upon mast cell activation.**
**(A)** Expression of *Zc3h12a-d* in bone marrow–derived mast cells (BMMCs) at the resting state (left panel) and upon activation with IgE and antigen complexes for up to 24 h (right), normalized to *Tbp* expression. N = 4–9 independent experiments. Mean ± SD. One-way ANOVA. **(B)** Expression of *Zc3h12a-d* in ex vivo–isolated peritoneal cavity mast cells at the resting state (left) and upon activation with PMA and ionomycin for 2 h (right), normalized to *Tbp* expression. N = 4–5 independent experiments. Mean ± SD. Paired *t* test, two-tailed. **(C)** Expression of Regnase-1 and Regnase-3 in BMMCs activated with IgE and antigen complexes or PMA and ionomycin at various time points. Regnase-1 (full-length + cleaved) and Regnase-3 (full-length) levels were normalized to *β*-tubulin expression. N = 3–4 independent experiments. Mean ± SD. One-way ANOVA. **(D)** Expression of Regnase-1 and Regnase-3 in cultured peritoneal cavity–derived mast cells activated with IgE and antigen complexes or PMA and ionomycin for 2 h, normalized to *β*-tubulin expression. N = 3 independent experiments. Mean ± SD. n.d., not determined. **(E)** Expression of *Zc3h12a* and *Zc3h12c* upon activation of BMMCs with different stimuli for 2 h, normalized to *Tbp* expression. N = 3–6 independent experiments. Mean ± SD. Paired *t* test, two-tailed.
Source data are available for this figure.

## Regnase-1 and -3 restrain the expression of inflammatory cytokines in mast cells

To gain a broad understanding of the impact of Regnase-1 and -3 on the expression inflammatory transcripts, we first used siRNAs to deplete both factors (schematics of siRNA targeting strategy in Fig 2A). Depletion of both Regnase-1 and -3, alone and in combination, was highly effective at the mRNA and protein levels (Fig 2B and C). Notably, depletion of Regnase-3 led to the significantly increased expression of Regnase-1, pointing toward a role of Regnase-3 in restraining Regnase-1 expression (Fig 2C), consistent with previous data described in macrophages (Liu et al, 2021; von Gamm et al, 2019). Next, we stimulated the cells for 2 h with IgE and antigen, and we measured the expression of 734 immune transcripts by NanoString digital profiling. We chose this method because we were primarily interested in immune-related inflammatory transcripts. We found that 325 of these transcripts were expressed by mast cells and that the depletion of Regnase-1, alone or in combination with

Regnase-3, led to the increased expression of mRNAs encoding inflammatory mediators, including established Regnase-1 targets such as *Nfkbiz* and *Il2* (Uehata et al, 2013; Jeltsch et al, 2014) (Fig 2D and Table S2). Differently from Regnase-1, depletion of Regnase-3 alone had only a very modest impact on the inflammatory transcriptome of mast cells, with only nine genes significantly up-regulated (log2 fold change >0.5 and *P* ≤ 0.05) (Fig 2D). Depletion of both proteins at the same time led to an overall combined phenotype that enhanced the effect of the individual depletions, with transcripts such as *Tnf, Il1b*, *Il2*, and *Nfkbiz* becoming even more prominently dysregulated (Fig 2D), suggesting redundant functions at least on specific targets. Overall, we found that Regnase-1, and to a lesser extent Regnase-3, restrains the expression of inflammatory transcripts in mast cells. The down-regulated transcripts were likely to be the result of indirect effects of Regnase depletion, and no previously described Regnase-bound, direct targets (Mino et al, 2015) were identified among the down-regulated transcripts (Fig 2E). Vice versa, among

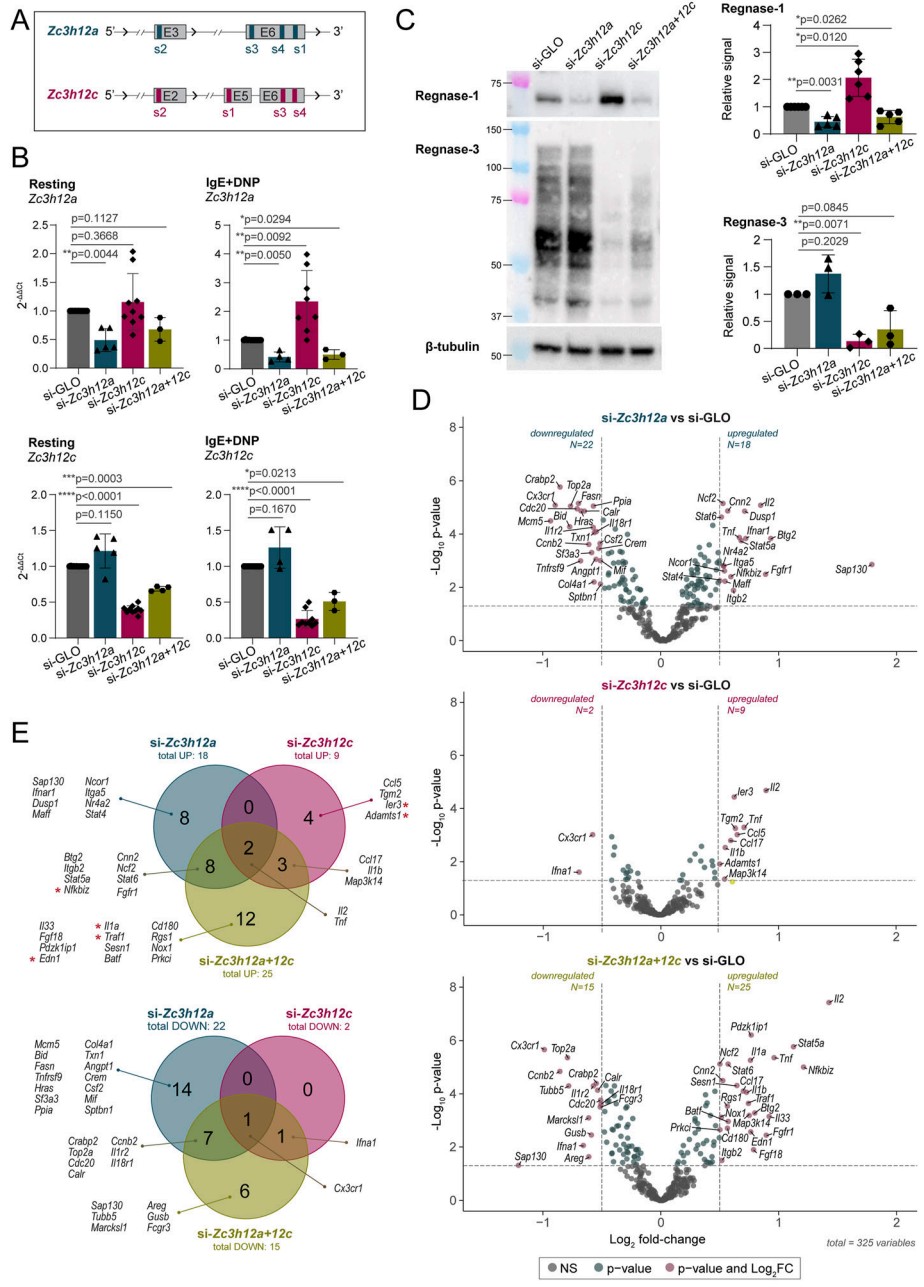

**Figure 2. Depletion of Regnase-1 and Regnase-3 induces inflammatory gene expression in activated mast cells.**
**(A)** Schematic representation of *Zc3h12a* and *Zc3h12c* loci with regions targeted by siRNAs for RNAi-mediated gene knockdown. Only the exons targeted by the siRNAs are shown. **(B, C)** mRNA and (C) protein expression of Regnase-1 and Regnase-3 upon transfection of bone marrow–derived mast cells (BMMCs) with siRNAs targeting *Zc3h12a* and/or *Zc3h12c*. Cells were either left resting or stimulated with IgE and antigen. mRNA expression is normalized to *Tbp* (B), whereas protein expression is normalized to $\beta$-tubulin (C). N = 3–9 independent experiments. Mean ± SD. Paired *t* test, two-tailed. **(D)** Differentially expressed genes (−0.5>log₂ fold change>0.5) in Regnase-1 and/or Regnase-3 knockdown BMMCs activated with IgE and antigen complexes for 2 h, measured by NanoString digital profiling. **(E)** Venn diagram of differentially expressed genes in (D). Genes marked with a red asterisk were reported to be directly bound by Regnase-1 in a RIP-seq analysis of HeLa cells (Mino et al, 2015).
Source data are available for this figure.

the significant up-regulated genes several direct Regnase-1 targets could be observed (red asterisks, Fig 2E), suggesting that our analysis identified transcripts at least in part directly modulated by Regnase proteins. Interestingly, *Il2* and *Tnf* were among the few inflammatory transcripts significantly increased upon depletion of both Regnase-1 and -3, alone or in combination. Because *Il2* was expressed at low levels in mast cells, we further investigated the impact of Regnases on TNF expression. First, we confirmed the increased expression of the *Tnf* transcript in an independent set of experiments (Fig 3A). Next, we found that, concordant with the mRNA data, TNF protein expression was significantly increased upon depletion of Regnase-3, alone or in

combination with Regnase-1, whereas the impact of Regnase-1 depletion alone was modest (Fig 3B). Next, to confirm these findings using an independent experimental setup, we used CRISPR/Cas9-mediated gene editing to deplete the expression of *Zc3h12a* or *Zc3h12c* (Fig 3C). Depletion of either protein was highly effective in mast cells, as shown by Western blot (Fig 3D). Concordant with our siRNA data, depletion of both Regnase-1 and -3 led to significantly increased *Tnf* mRNA expression (Fig 3E) and to increased protein expression, measured by intracellular staining, both in response to IgE and antigen complexes or PMA and ionomycin stimulation (Fig 3F). We also measured the expression of other cytokines prominently produced by mast cells. We found

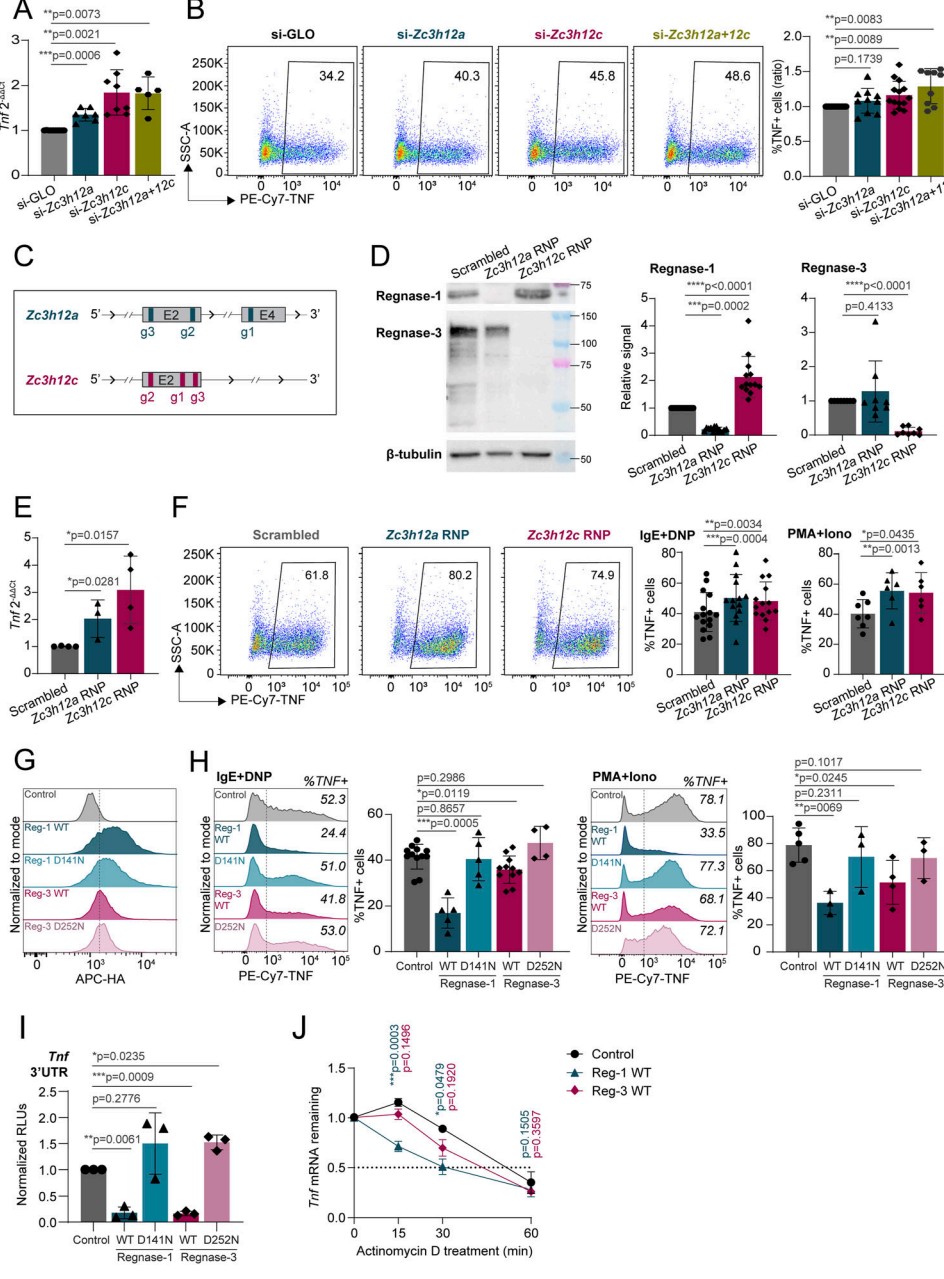

**Figure 3. Regnase-1 and Regnase-3 negatively regulate TNF expression.**
**(A, B)** mRNA expression and (B) intracellular staining of TNF in activated bone marrow–derived mast cells (BMMCs) (IgE and antigen complexes for 2 h in (A) and 4 h in (B)) transfected with siRNAs against *Zc3h12a* and/or *Zc3h12c*. mRNA expression is normalized to *Tbp* in (A). Representative FACS plots and percentage of TNF+ cells (normalized to siGLO) are shown in (B). N = 5–14 independent experiments. Mean ± SD. Paired *t* test, two-tailed. **(C)** Schematic representation of *Zc3h12a* and *Zc3h12c* loci with regions targeted by gRNAs for CRISPR/Cas9-mediated gene knockout. Only the exons targeted by the gRNAs are shown. **(D)** Expression of Regnase-1 and Regnase-3 upon transfection of BMMCs with CRISPR/Cas9-gRNA RNPs targeting *Zc3h12a* or *Zc3h12c* after 1 wk of culture, normalized to *β*-tubulin expression. N = 8–13 independent experiments. Mean ± SD. Paired *t* test, two-tailed. **(E)** *Tnf* expression in Regnase-1 and Regnase-3 knockout BMMCs activated with IgE and antigen complexes for 2 h, normalized to *Tbp* expression. N = 3–4 independent experiments. Mean ± SD. Unpaired *t* test, two-tailed. **(F)** Intracellular staining of TNF in Regnase-1 and -3 knockout cells activated with IgE and antigen complexes (left) or PMA and ionomycin (right) for 4 h. N = 6–14 independent experiments. Mean ± SD. Paired *t* test, two-tailed. **(G)** Intracellular staining of HA-tag in BMMCs transfected with in vitro–transcribed HA-tagged *Zc3h12a* or *Zc3h12c* mRNA to assess the overexpression of Regnase-1 and -3, either WT or RNase-inactive mutant (D141N or D252N). **(H)** Intracellular staining of TNF in Regnase-1– and -3–overexpressing cells activated with IgE and antigen complexes (left) or PMA and ionomycin (right) for 4 h. N = 3–11 independent experiments. Mean ± SD. Paired *t* test, two-tailed. **(I)** Luciferase reporter assay upon co-transfection of Regnase-1 or Regnase-3, either WT or RNase-inactive mutant (D141N and D252N), with a *Tnf* 3′UTR reporter plasmid in HEK293T cells. N = 3 independent experiments. Mean ± SD. Paired *t* test, two-tailed. **(J)** BMMCs were transfected with in vitro–transcribed mRNAs encoding Regnase-1 or Regnase-3, followed by IgE stimulation and actinomycin D treatment. *Tnf* expression was measured over time by RT–qPCR. N = 4 independent experiments. Mean ± SEM. Paired *t* test, two-tailed. Source data are available for this figure.

that IL-6 expression was modestly affected by Regnase-1 deletion (Fig S4A), consistent with our NanoString data, showing mildly increased expression, and it was unaffected by Regnase-3 deletion. Although at the mRNA level we could not detect any effect on the *Il13* transcript, IL-13 expression was mostly increased upon deletion of either Regnase, suggesting indirect regulation.

To confirm these data, we sought to force the expression of Regnase enzymes in mast cells. However, we found that most overexpression methods, including inducible vectors, failed to

provide stable Regnase-1 and -3 protein expression, because of deleterious effects, especially of Regnase-1 on mast cell viability. We therefore optimized an in vitro transcription (IVT) system for the direct, short-term delivery of mRNAs encoding Regnase-1 and -3. The T7-driven transcripts were stabilized by the addition of the 5′UTR and 3′UTR from the *HBB* gene, followed by 5′ capping and 3′ poly(A) tailing. We generated transcripts leading to the expression of HA-tagged WT Regnase-1 and -3 and versions carrying point mutations in the RNase enzymatic domain (D141N and D252N, respectively), leading to disruption of the catalytic activity

(Matsushita et al, 2009). mRNA delivery of Regnase-1 and -3, either WT or RNase mutant, proved to be feasible and efficient, and all proteins were highly expressed in transfected mast cells, as measured by intracellular staining using an antibody against the HA-tag (Figs 3G and S4B). Cell viability was comparable across conditions (Fig S4B), and the overall subcellular localization of the IVT mRNA-derived proteins was comparable to that of endogenously expressed Regnase-1 and -3 (Fig S5). Analysis of cytokine expression by these cells upon stimulation with either IgE and antigen complexes or PMA and ionomycin revealed that both Regnase-1 and -3 significantly diminished the ability of mast cells to produce TNF in an RNase-dependent manner (Fig 3H). Similar results were obtained by measuring TNF release in the culture supernatant by ELISA (Fig S4C). Compared with Regnase-1, the somewhat reduced capability of Regnase-3 to limit TNF expression was likely linked to the extent of overexpression that we could achieve in this experimental system, which was less pronounced for Regnase-3 (Fig 3G). Mostly consistent with our CRISPR/Cas9 knockout data, the overexpression of Regnase-3 had no detectable effects on IL-6 production, whereas Regnase-1 limited both IL-6 and IL-13 expression by mast cells (Fig S6A). Next, we used the human mast cell lines HMC-1.1 and HMC-1.2 (Sundström et al, 2003) to assess the impact of Regnase-1 on TNF expression in the human system. We found that IVT transfection of human Regnase-1, but not a catalytically inactive version (D141N), significantly reduced TNF expression by human mast cells (Fig S6B and C), indicating that Regnase-dependent regulation of TNF expression is relevant also in the context of human mast cells. To determine whether the effect of Regnase-1 and -3 on murine TNF expression was mediated by the direct targeting of the *Tnf* transcript, we cloned the full-length 3′UTR of *Tnf* into a luciferase reporter vector. We then co-transfected it with Regnase-1– or Regnase-3–expressing vectors, either WT or RNase-impaired. We found that the *Tnf* 3′UTR was strongly affected by both Regnases and that luciferase expression was restored upon mutation of their RNase enzymatic sites, pointing toward a direct effect on *Tnf* that required the RNase enzymatic activity of Regnase proteins (Fig 3I). Finally, actinomycin D treatment of cells transfected with IVT mRNA encoding Regnase-1 or Regnase-3 showed reduced stability over time of the *Tnf* mRNA (Fig 3J). Again, this effect was more prominent for Regnase-1, and potentially linked to the higher level of expression that we could achieve in the IVT transfection system, compared with Regnase-3. Overall, both Regnase-1 and -3 are required to effectively limit TNF expression by mast cells, although their effect on other inflammatory cytokines appears to be more selective and predominantly linked to Regnase-1.

### Regnase-3 is required to restrain the expression of Regnase-1 in mast cells

Depletion of Regnase-3 by RNAi led to significantly increased Regnase-1 expression in mast cells (Fig 2B and C), indicating that understanding the phenotype of either knockdown or knockout is complicated by the presence of direct and indirect effects and cross-regulation between these two factors. To better dissect the role of Regnase-3 in regulating Regnase-1 expression, we first confirmed the effect of Regnase-3 on the *Zc3h12a* mRNA by deleting

*Zc3h12c* by CRISPR/Cas9, showing that *Zc3h12a* mRNA expression was indeed significantly increased (Fig 4A). Concordant with this result, we found that upon overexpression of Regnase-3 by IVT mRNA transfection, Regnase-1 protein expression was significantly reduced (Fig 4B). Regnase-1 is also known to diminish the stability of its own transcript through stem–loop structures in the 3′UTR (Iwasaki et al, 2011; Tse et al, 2022). To assess the impact of the individual Regnases on *Zc3h12a* expression, we cloned the 3′UTR of *Zc3h12a* in a luciferase reporter plasmid and measured its response upon co-transfection of Regnase-1 or Regnase-3. Both proteins were able to strongly affect luciferase expression in an RNase activity–dependent manner, suggesting a direct effect on the 3′UTR of *Zc3h12a* (Fig 4C). Finally, to determine whether Regnase-3 was required to modulate *Zc3h12a* transcript stability in mast cells, we treated the cells with actinomycin D, followed by the measurement of *Zc3h12a* mRNA levels over time. Deletion of Regnase-3 was sufficient to increase the half-life of the *Zc3h12a* transcript in mast cells (Fig 4D), whereas the overexpression of Regnase-3 by IVT mRNA transfection reduced it (Fig 4E). On the contrary, we found no evidence of any impact of Regnase-1 depletion or deletion on Regnase-3 (Fig 4F).

To gain more insights into the crosstalk between Regnase-1 and -3, we performed immunoprecipitation and mass spectrometry analyses of the endogenously expressed proteins in mast cells activated with PMA and ionomycin (Figs 4G and H and S7A, and Table S3). We found that immunoprecipitation of Regnase-1 recovered Regnase-3 as its primary endogenous interactor (Fig 4G). This interaction was confirmed in HEK293T cells upon co-transfection of Regnase-1 and Regnase-3 (Figs 4I and S7B), and in BMMCs sequentially transduced to overexpress catalytically inactive Regnase-1 D141N and Regnase-3 D252N (Figs 4J and S7C). Similar results were obtained using an anti-FLAG antibody to immunoprecipitate FLAG-tagged Regnase-1 or Regnase-3 (Fig S7D). Furthermore, the interaction between Regnase-1 and -3 was unaffected by RNase treatment, suggesting that it does not require RNA binding to occur (Fig S7E). On the contrary, Regnase-3 interacted primarily with ribosomal proteins, suggesting an involvement in translation, and with the 14-3-3 family of proteins (Fig 4H). 14-3-3 proteins were previously shown to interact with FLAG-HA-tagged Regnase-1 when transfected into HeLa cells and to contribute to Regnase-1 protein stability (Akaki et al, 2021). Because in that system Regnase-3 was not expressed, it could not be recovered as an interactor of Regnase-1. However, in our experimental setting in which both Regnase family members are expressed, we found that complex formation involves primarily Regnase-1 and -3 on the one side and Regnase-3 and 14-3-3 proteins on the other, although cell- and stimulus-specific differences may also apply. The fact that in mast cells Regnase-3 was identified as an interactor of Regnase-1, but not the opposite, suggests that Regnase-3 is predominantly interacting with other protein partners, and only a relatively small proportion remains available for Regnase-1 interaction. To determine which region(s) of the Regnase-3 protein are important for its interaction with Regnase-1, we generated truncation mutants lacking different domains of Regnase-3 (Fig 4K). We then co-transfected HEK293T cells and confirmed protein expression by immunoblot (Fig 4L). Next, immunoprecipitation of Regnase-1 recovered Regnase-3 as an interacting partner of all Regnase-3 constructs except for the shortest (1–245aa) protein, lacking the PIN domain (Fig 4M).

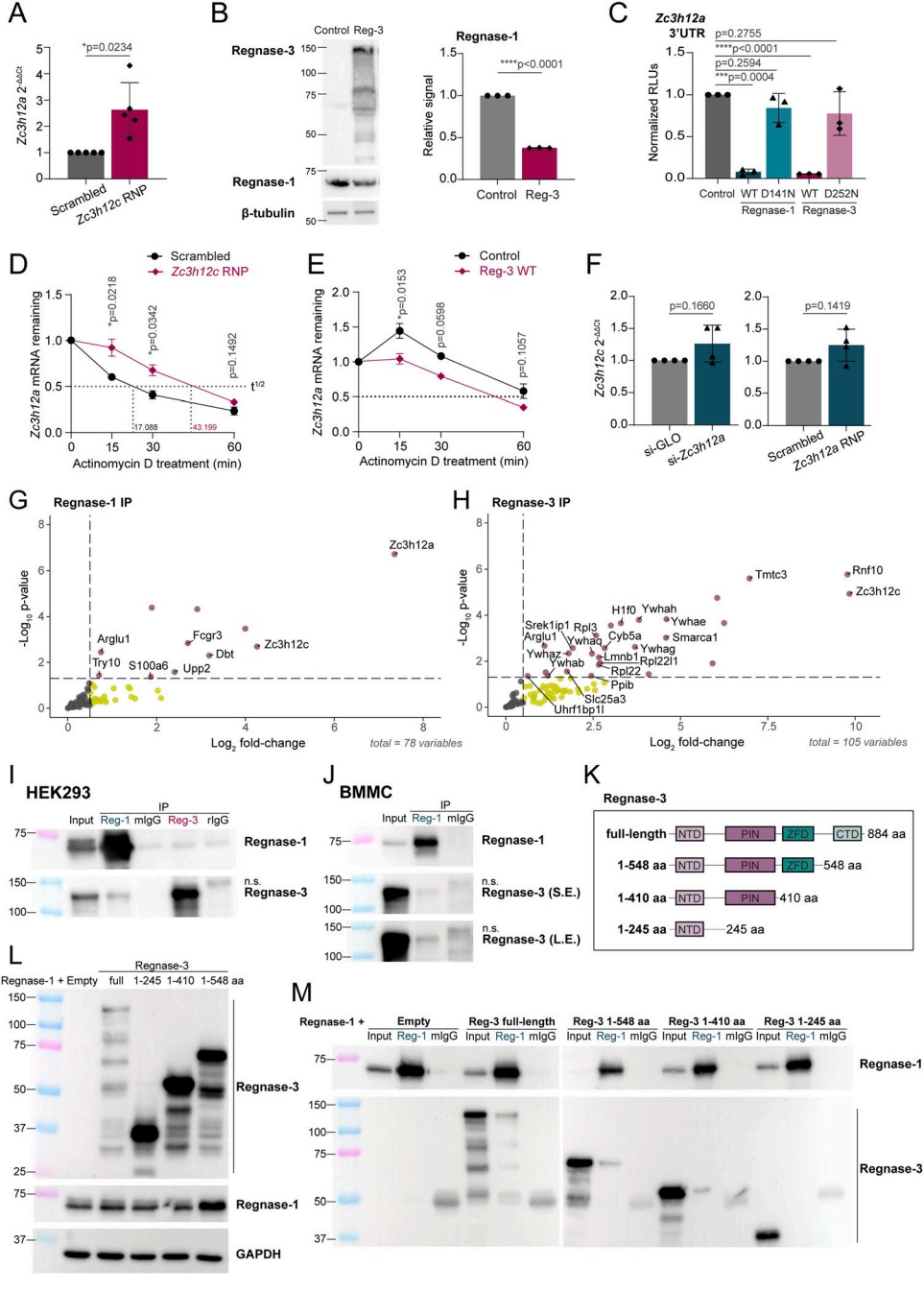

**Figure 4. Regnase-3 acts as a negative regulator of Regnase-1 by destabilizing the *Zc3h12a* transcript.**
**(A)** Expression of *Zc3h12a* in Regnase-3 knockout bone marrow–derived mast cells (BMMCs) activated with IgE and antigen complexes for 2 h, normalized to *Tbp*. N = 5 independent experiments. Mean ± SD. Ratio-paired *t* test, two-tailed. **(B)** BMMCs were transfected with in vitro–transcribed *Zc3h12c* mRNA, followed by Western blot to assess the effect of Regnase-3 overexpression on Regnase-1 expression, normalized to *β*-tubulin. N = 3 independent experiments. Mean ± SD. Paired *t* test, two-tailed. **(C)** HEK293T cells were transfected with plasmids expressing Regnase-1 or Regnase-3, either WT or RNase-inactive mutant (D141N and D252N), together with a *Zc3h12a* 3′UTR reporter plasmid, followed by luciferase assay. N = 3 independent experiments. Mean ± SD. Paired *t* test, two-tailed. **(D)** BMMCs were transfected with *Zc3h12c* RNPs (or scrambled RNP control), followed by actinomycin D treatment to block transcription. *Zc3h12a* transcript stability was measured by RT–qPCR at the indicated time points after stimulation with IgE and antigen complexes. N = 4 independent experiments. Mean ± SEM. Paired *t* test, two-tailed. Half-life (t^1/2) was calculated using non-linear regression analysis. **(E)** BMMCs were transfected with in vitro–transcribed mRNA encoding Regnase-3, followed by IgE stimulation and actinomycin D treatment. The expression of *Zc3h12a* was measured at the indicated time points by RT–qPCR. N = 4 independent experiments. Mean ± SEM. Paired *t* test, two-tailed. **(F)** Expression of *Zc3h12c* in Regnase-1 knockdown (left) or knockout (right) BMMCs activated for 2 h with IgE and antigen complexes, normalized to *Tbp*. N = 4 independent experiments. Mean ± SD. Ratio-paired *t* test, two-tailed. **(G, H)** Immunoprecipitation and mass spectrometry analyses of endogenous (G) Regnase-1 and (H) Regnase-3 in BMMCs activated with PMA and ionomycin for 1 h. Immunoprecipitation with mouse or rat IgG isotype control was used as control. Proteins with log₂FC > 0 from 3 to 4 independent experiments are shown. **(I)** Western blot analysis upon immunoprecipitation of Regnase-1, Regnase-3, and corresponding isotype control in HEK293T cells co-transfected with Regnase-1 and Regnase-3 for 48 h. mIgG, mouse IgG; rIgG, rat IgG; n.s., non-specific. **(J)** BMMCs were sequentially transduced with lentiviruses encoding the catalytically inactive Regnase-1 D141N followed by Regnase-3 D252N. Immunoprecipitation was performed using an anti-Regnase-1 antibody, followed by immunoblot using an anti-Regnase-3 antibody. S.E., short exposure; L.E., long exposure. **(K)** Schematic diagram of Regnase-3 truncation mutants. NTD, N-terminal domain; PIN, PilT N-terminus RNase domain; ZFD, zinc finger domain; CTD, C-terminal domain. **(L)** HEK293T cells were co-transfected with plasmids encoding Regnase-1 and Regnase-3 (both full-length and truncation mutants), followed by immunoblot to detect the expression of both proteins. **(M)** Cells as in (L) were used for immunoprecipitation of Regnase-1, followed by immunoblot to detect Regnase-3 co-immunoprecipitation. Representative of two independent experiments.
Source data are available for this figure.

This is consistent with the PIN domain being important for protein oligomerization, as shown previously for Regnase-1 (Yokogawa et al, 2016). In sum, our data reveal that Regnase-3 is a direct interactor and regulator of Regnase-1 expression in mast cells.

## Regnase-1, but not Regnase-3, is required to maintain mast cell proliferation and viability

Having established the role of Regnase-1 and -3 in regulating inflammatory responses in mast cells, we investigated their role in

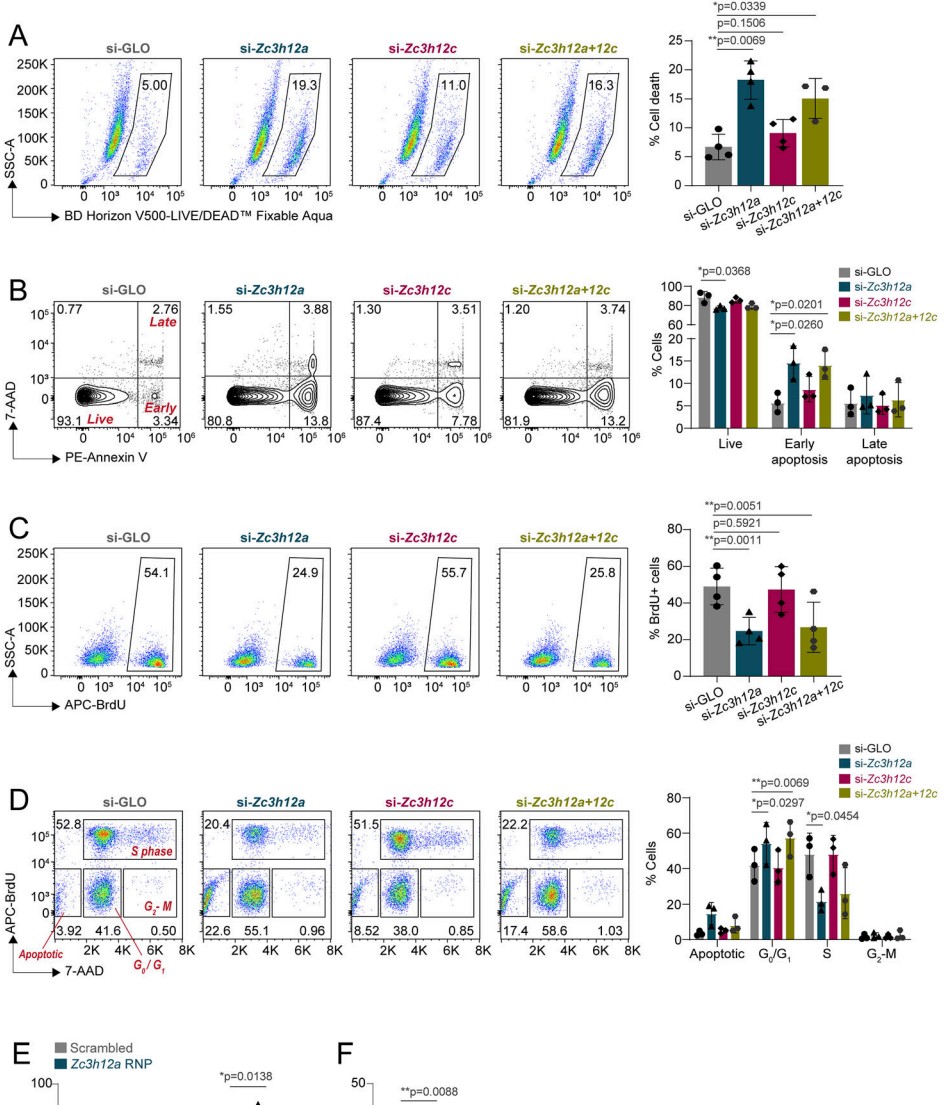

**Figure 5. Regnase-1 is crucial in maintaining mast cell homeostatic functions.**

**(A, B, C, D)** Bone marrow–derived mast cells (BMMCs) were transfected with siRNAs targeting the indicated Regnase family members, followed by phenotypic analyses 72 h after transfection. **(A)** LIVE/DEAD staining to measure cell viability. N = 3–4 independent experiments. Mean ± SD. Paired *t* test, two-tailed. **(B)** Annexin V and 7-AAD staining to measure cell apoptosis. N = 3 independent experiments. Mean ± SD. Unpaired *t* test, two-tailed. **(C)** BrdU staining to measure cell proliferation. N = 4 independent experiments. Mean ± SD. Paired *t* test, two-tailed. **(D)** BrdU and 7-AAD co-staining to measure cell cycle progression. N = 3 independent experiments. Mean ± SD. Two-way ANOVA. **(E, F)** BMMCs were transfected with Cas9/gRNA RNPs targeting *Zc3h12a*. The following experiments were performed after 1 wk. **(E)** LIVE/DEAD staining to measure cell viability after growing cells in medium with or without IL-3 for 72–96 h. N = 3–4 independent experiments. Mean ± SD. Paired *t* test, two-tailed. **(F)** BrdU staining to measure cell proliferation. N = 3 independent experiments. Mean ± SD. Paired *t* test.

Source data are available for this figure.

modulating mast cell growth and survival. First, depletion of Regnase-1, but not Regnase-3, was sufficient to reduce cell viability 72 h after siRNA transfection, measured by LIVE/DEAD staining (Fig 5A). Consistent with this observation, measuring the extent of apoptosis by Annexin V staining revealed a significant increase in the percentage of early apoptotic cells upon Regnase-1 depletion (Fig 5B), as well as diminished ability of the cells to proliferate, measured by BrdU incorporation (Fig 5C). Cell cycle analysis confirmed an increased percentage of cells in $G_1$ and a concomitant decrease in the percentage of cells in the S phase (Fig 5D). A similar increase in cell death was observed upon deletion (by CRISPR/Cas9) of *Zc3h12a*, which remained significant regardless of the presence of IL-3 (an essential survival factor for mast cells) in the culture medium (Figs 5E and S8A) and was accompanied by reduced overall proliferation (Figs 5F and S8B). Meanwhile, deletion of Regnase-1 did not affect the capacity of the cells to degranulate in response to IgE and antigen stimulation (Fig S8C).

Because the limited gene set in the NanoString panel provided mainly information relative to inflammatory genes, to gain insights into the mechanism(s) that may lead to such pronounced impairment in the ability of mast cells to survive in the absence of Regnase-1, we deleted Regnase-1 by CRISPR/Cas9 and performed RNA sequencing. We found that 131 genes were up-regulated, whereas 187 genes were down-regulated in the absence of

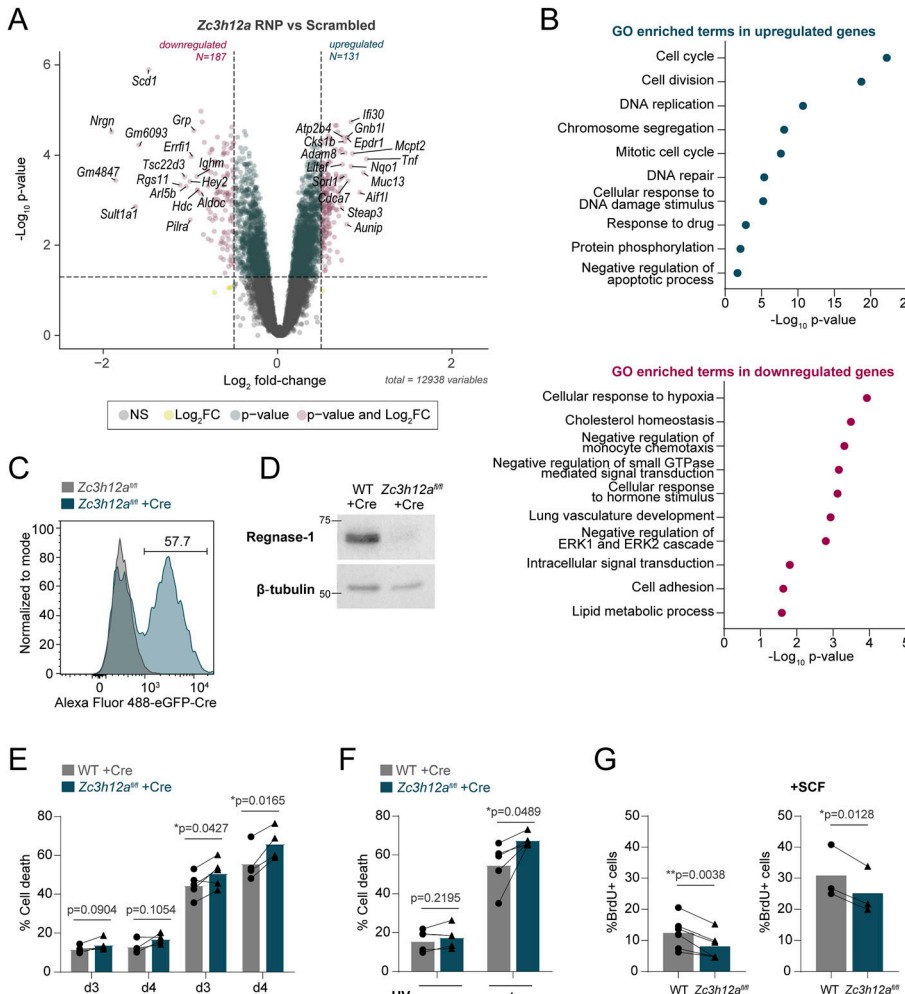

**Figure 6. Loss of Regnase-1 induces widespread transcriptome changes in mast cells.**
**(A)** Differential gene expression in bone marrow–derived mast cells (BMMCs) transfected with scrambled (control) or *Zc3h12a*-targeting Cas9-gRNA RNPs at the resting state, measured by RNA-seq. **(B)** Gene ontology analysis of enriched terms in up-regulated and down-regulated genes in (A). **(C)** Representative histogram of EGFP expression in WT and *Zc3h12a*^fl/fl BMMCs transduced with an EGFP-Cre recombinase–expressing lentivirus. Subsequent experiments were performed 4–7 d after transduction, gating on EGFP-positive cells. **(D)** Expression of Regnase-1 in BMMCs differentiated from WT or *Zc3h12a*^fl/fl mice after 1 wk of transduction with an EGFP-Cre recombinase–expressing lentivirus, representative of N = 4 independent experiments. **(E)** LIVE/DEAD staining to measure cell viability after growing cells in medium with or without IL-3 for 72–96 h. N = 4–5 independent experiments. Mean, paired *t* test, two-tailed. **(F)** Cell viability measured by LIVE/DEAD staining 72 h after UV treatment (200 J/m²). N = 5 independent experiments. Mean, paired *t* test, two-tailed. **(G)** BrdU staining to measure cell proliferation in normal culture medium containing IL-3 or supplemented with stem cell factor. N = 3–6 independent experiments. Mean, paired *t* test, two-tailed.
Source data are available for this figure.

Regnase-1, pointing toward both direct and indirect effects (Fig 6A and Table S4). Among the up-regulated genes, the basal expression of *Tnf* was increased, even in the absence of stimulation. Some previously reported direct targets of Regnase-1 (Mino et al, 2015), including *Nfkbid* and *Ran*, were also modestly up-regulated (Table S4). Transcripts encoding the mast cell protease Mcpt2 and the metalloprotease Adam8 were increased, whereas some genes involved in regulating mast cell degranulation such as *Hdc* (Nakazawa et al, 2014) were down-regulated. GO analysis of the down-regulated genes clustered around signal transduction, as well as cholesterol and lipid metabolism (Fig 6B). These genes include *Scd1*, *Scd2*, *Acsl3*, and *Fads2*, which are important for lipid synthesis, and *Abcg1*, *Abca1*, *Pcsk9*, and *Fabp5*, which are involved in lipid transport, pointing toward overall dysregulation in lipid homeostasis. Metabolism-related genes downmodulated in the absence of Regnase-1 also included *Aldoc*, *Pdk1*, and *Hk2*, which play a role in glucose metabolism. These results point toward involvement of Regnase-1 in regulating metabolic processes required for cell maintenance, as observed also in other cell types (Younce et al, 2009; Nagahama et al, 2018; Behrens et al, 2021; Reina-Campos et al, 2023). On the contrary, categories associated with the up-regulated

genes revealed many GO terms associated with the control of cell cycle, and DNA replication and repair (Fig 6B). Indeed, some of the most differentially expressed genes included factors important for cell cycle progression, such as *Cdca7*, *Cks1b*, and *Cdk1*. Although classical anti- and pro-apoptosis–related genes, such as *Bcl2*, *Bax*, and *Bcl2l1*, were unchanged, DNA damage and replication markers such as *Aunip*, *Ung*, *Pold1*, and *H2ax* were up-regulated, suggesting that Regnase-1 knockout cells were more susceptible to cellular stress, in addition to cell cycle defects.

To verify this phenotype using an independent experimental approach, we took advantage of mast cells derived from the bone marrow of *Zc3h12a*^fl/fl mice (Li et al, 2017). Briefly, we transduced bone marrow cells with a lentiviral vector expressing an EGFP-Cre fusion protein (Leoni et al, 2023) reaching at least 50% efficiency of transduction (Fig 6C). After sorting of EGFP⁺ cells, the expression of Regnase-1 became undetectable by Western blot (Fig 6D). Similar to cells treated with RNAi or Cas9 RNPs, *Zc3h12a*^fl/fl-Cre mast cells showed reduced viability in different stress conditions, including the withdrawal of the survival factor IL-3 and short-term UV irradiation (Figs 6E and F and S8D and E). Proliferation was similarly reduced in these cells even in conditions that optimally sustain

mast cell growth (addition of IL-3, with or without SCF) (Figs 6G and S8F). Overall, using numerous independent experimental approaches, we found that Regnase-1 is central to the maintenance of mast cell survival and basal proliferation, and that Regnase-1 deletion leads to widespread transcriptome changes associated with cell cycle defects.

# Discussion

In this study, we found that Regnase-1 and Regnase-3 contribute to the regulation of mast cell responses and homeostatic activities. Our data extend and complement previous studies showing negative regulation of inflammatory type 2 responses by Regnase-1 in other cell types. For instance, Regnase-1 was shown to modulate the activity of group 2 innate lymphoid cells, and the inhibition of Regnase-1 degradation in these cells led to attenuated pulmonary inflammation (Matsushita et al, 2020). Similarly, deletion of Regnase-1 in Th2 lymphocytes led to enhanced IL-5 production by these cells and lung inflammation (Peng et al, 2018). Compared with Regnase-1, current knowledge regarding the impact of Regnase-3 in modulating type 2 responses is more limited. Regnase-3 was primarily shown to modulate macrophage and dendritic cell responses, where it was shown to affect cytokine production, including TNF and IL-6 (Liu et al, 2021), although one of the primary roles of Regnase-3 in macrophages was shown to be the regulation of Regnase-1 expression (von Gamm et al, 2019). We now showed that in mast cells, the regulation of responses to IgE crosslinking requires the cooperative function of both Regnase-1 and Regnase-3. Among the transcripts differentially expressed upon Regnase-1 and -3 silencing, *Tnf* was found to be regulated by both proteins through their intrinsic enzymatic activities and by the direct targeting of the *Tnf* 3′UTR. An additive effect was clearly observed upon co-depletion of both Regnases, although differences in dosage or potency of RNase activity cannot be ruled out.

Post-transcriptional regulation of *Tnf* through the AU-rich elements (ARE) within its 3′UTR was previously described (Kontoyiannis et al, 1999) and mostly attributed to the action of ARE-binding proteins, such as tristetraprolin (TTP) (Lai et al, 2000; Suzuki et al, 2003). ARE-independent regulation through other regulatory elements was also reported, including a constitutive decay element recognized by Roquin-1/2 (Stoecklin et al, 2003; Leppek et al, 2013) and a new regulatory element shown to be targeted by Regnase proteins, at least in reporter assays (Lacey et al, 2015). Among these elements, removal of the ARE and new regulatory element led to strong derepression of TNF expression and to embryonic lethality in mice (Lacey et al, 2015; Clayer et al, 2020). In mast cells, TNF was previously reported to be regulated by TTP in an ARE-dependent manner in response to IL-4 (Suzuki et al, 2003), although not in response to stimulation with LPS (Hochdörfer et al, 2013), exemplifying context-dependent post-transcriptional regulation of this cytokine. Our data now provide evidence of the existence of an additional mechanism of TNF regulation exerted in mast cells by Regnase-1 and Regnase-3 in response to IgE stimulation. Regarding the specific effects of Regnase-1 or Regnase-3 on the *Tnf* 3′UTR, at least two scenarios can be envisioned. One possibility is that both

proteins have exactly the same function, and any phenotypic difference is due to differences in their level of expression and subcellular localization. A second possibility is instead linked to differences in binding affinity and activity between the two proteins. These may be dependent on co-factors and post-translational modifications that can be at least in part signal- and cell type–specific. Importantly, we found that Regnase-3 is capable of stable interactions with Regnase-1, which may be important for their shared regulation of common inflammatory targets, such as *Tnf*. In previous studies, the interaction of Regnase-1 with other proteins, such as UPF1 and Roquin-1 (Mino et al, 2015, 2019; Behrens et al, 2021), was shown to enhance its activity and specificity, whereas proteins involved in Regnase-1 post-translational modifications (MALT-1, IKKs/IRAK1, 14-3-3) influenced its stability and subcellular localization (Iwasaki et al, 2011; Uehata et al, 2013; Jeltsch et al, 2014; Akaki et al, 2021). The protein interactors of Regnase-3, on the contrary, were so far largely unknown. Compared with Regnase-1, Regnase-3 not only shares homology in the PIN and zinc finger domains (von Gamm et al, 2019), but also contains a long non-homologous C-terminal region that could serve as a hub for protein interactions that may endow Regnase-3 with unique roles. Indeed, immunoprecipitation of endogenous Regnase-3 in mast cells revealed unique protein interactions that hint at its potential role in translation processes.

In terms of gene expression, we found that the basal expression of *Zc3h12a* and *Zc3h12c* was comparable in resting mast cells, but the extent of induction upon IgE-dependent activation was higher for *Zc3h12c* (~30-fold after 2 h, compared with a maximal induction for *Zc3h12a* of ~threefold). A similar, larger magnitude of induction of Regnase-3 compared with Regnase-1 was also observed at the protein level (~fourfold versus ~twofold, respectively). One possible interpretation of these findings is that Regnase-1 is primarily linked to the regulation of resting-state transcripts, whereas stimulus-induced inflammatory transcripts can be regulated by both proteins in a dose-dependent manner. Consistent with this possibility, we found that both Regnase-1 and -3 modulated the *Tnf* 3′UTR to a similar extent. However, the observation that these two proteins interact with each other but also with unique interactors raises the possibility that in intact cells, the two proteins have indeed regulatory functions that are not fully redundant. Accordingly, HITS-CLIP analyses in transfected HEK293T cells showed an incomplete overlap in their target binding specificity (Uehata et al, 2024). The fact that we found very limited co-localization of Regnase-1 and -3 by immunofluorescence even in overexpressing cells suggests that part of the incomplete target overlap may be linked to the physical segregation of these two proteins. The narrow influence of Regnase-3 on inflammatory gene expression may also be in part due to compensation by Regnase-1. Indeed, Regnase-3 negatively regulates Regnase-1 expression by destabilizing the *Zc3h12a* transcript through its RNase activity.

Consistent with a role of both Regnase-1 and Regnase-3 in co-regulating at least some stimulus-induced responses in mast cells, and with a predominant role of Regnase-1 in resting functions, concomitant knockdown of both proteins revealed mostly additive effects on inflammatory gene expression, suggesting cooperative and possibly redundant functions in restraining inflammation. However, in resting mast cells, only Regnase-1 was required for

steady-state cell survival and proliferation. Regnase-3 knockdown did not exhibit an opposite phenotype, nor did the double knockdown result in any additive or antagonistic effects with respect to Regnase-1 knockdown alone. This implies that beyond dosage compensation, Regnase-1 may carry out unique roles in maintaining normal mast cell homeostasis. Indeed, RNA sequencing of mast cells lacking Regnase-1 revealed that many differentially expressed genes were associated with cell cycle and DNA replication, and with metabolic processes, including lipid and glucose metabolism. It is plausible that these phenotypes resulted at least in part from indirect effects of Regnase-1 deletion, because only a few of the established direct Regnase-1 targets were dysregulated. For example, the increased cell death observed in mast cells lacking Regnase-1 is likely the result of cell cycle defects and replication stress, and of the overall poor cellular fitness observed in the absence of Regnase-1.

Overall, our study revealed that Regnase-1 and -3 constitute a network that cooperatively fine-tunes inflammatory gene expression during mast cell activation. Regnase-1 is in addition critical for mast cell survival and proliferation, whereas Regnase-3 is a direct interactor of Regnase-1 and is required to restrain Regnase-1 expression. Although our study focused primarily on murine cells, given the high species conservation between human and mouse Regnase-1 and -3 proteins (82.5% sequence identity for Regnase-1, 92% sequence identity for Regnase-3), our results are likely to apply more broadly also to humans, as shown also by the fact that TNF expression appears to be similarly regulated by Regnase-1 in both human and mouse mast cells. In sum, we propose a model that distinguishes the role of Regnase-1 and -3 in resting and activated mast cells. In the resting state, Regnase-3 is lowly expressed, whereas Regnase-1 expression is required to maintain mast cell homeostatic proliferation and survival. Upon activation, both proteins are strongly and dynamically induced, and they contribute to restrain excessive inflammatory responses (most notably TNF production). Regnase-3 induction also contributes to limit Regnase-1 expression, potentially favoring the re-establishment of a post-activation, quiescent state.

# Materials and Methods

### Mice

All mice used in this study were on a C57BL/6 background, housed in a specific pathogen-free barrier facility under 12-h dark/light cycles, 20–24°C temperature, and 50–65% humidity conditions. All animal studies were performed in accordance with Swiss Federal Veterinary Office guidelines and with approval from the Cantonal animal experimentation committee, Dipartimento della Sanità e della Socialità Cantone Ticino (authorization number TI10/19).

### Cell cultures

BMMCs were in vitro–differentiated from the bone marrow of 6- to 8-wk-old mice. Whole bone marrow cells were cultured in IMDM (Gibco) containing 10% heat-inactivated FBS (Gibco) and IL-3 (recombinant, prepared in-house), and supplemented with 0.1 mM non-essential amino acids (Gibco), 2 mM L-alanyl-L-glutamine dipeptide (Gibco), 100 U/ml penicillin, 100 mg/ml streptomycin (Gibco), and 50 $\mu$M $\beta$-mercaptoethanol, as previously described (Montagner et al, 2016; Leoni et al, 2017). After 4 wk, ~99% of the cells were Fc$\varepsilon$RI$\alpha^+$ c-Kit$^+$ and were used for downstream analyses. For all experiments, BMMCs between week 4 and week 8 of age were used. Peritoneal cavity mast cells were isolated ex vivo from mouse peritoneal cavity by intraperitoneal lavage followed by magnetic bead–based enrichment for c-Kit$^+$ cells using anti-CD117 (c-Kit)-APC antibody (BioLegend) and APC microbeads (Miltenyi Biotec). Cells were expanded for at least 1 wk in the presence of IL-3 and 30 ng/ml recombinant SCF (Peprotech) to obtain peritoneal cavity–derived mast cells. To generate mast cells lacking Regnase-1, BMMCs from $Zc3h12a^{fl/fl}$ mice (Li et al, 2017) was transduced with a lentiviral vector expressing the Cre recombinase fused to an EGFP reporter (Leoni et al, 2023). Downstream analyses were performed by gating on EGFP$^+$ cells (for flow cytometry–based assays) or by sorting for EGFP$^+$ cells using BD FACSymphony S6 Cell Sorter (BD Biosciences) (for Western blot experiments). HEK293T cells were maintained in DMEM (Gibco) supplemented with 10% FBS (Gibco), 1 mM sodium pyruvate (Gibco), 100 U/ml penicillin, 100 mg/ml streptomycin (Gibco), and 50 $\mu$M $\beta$-mercaptoethanol. The HMC-1.1 and HMC-1.2 human mast cell lines were kindly provided by Joseph Butterfield and were cultured as described previously (Sundström et al, 2003).

### Mast cell activation

Murine mast cell cultures were stimulated with the following for the indicated time periods: 1 $\mu$g/ml IgE/anti-DNP antibody (Sigma-Aldrich) and 0.2 $\mu$g/ml HSA-DNP (Sigma-Aldrich), 20 nM PMA (Sigma-Aldrich) and 2 $\mu$M ionomycin (Sigma-Aldrich), 10 ng/ml recombinant IL-33 (BioLegend), 1 $\mu$g/ml LPS (InvivoGen), and 10 ng/ml SCF (Peprotech). Human HMC-1.1 and HMC-1.2 cells were stimulated with 20 nM PMA (Sigma-Aldrich) and 2 $\mu$M ionomycin (Sigma-Aldrich) for 3 h.

### Reverse transcription–quantitative PCR (RT–qPCR)

Mast cells were lysed in TRI Reagent RT (Molecular Research Center), and total RNA was extracted using Direct-zol RNA Micro-prep Kit (Zymo Research) according to the manufacturer's protocol. cDNA was synthesized by reverse transcription using qScript cDNA SuperMix (Quanta Bioscience). Target genes were amplified using PerfeCTa SYBR Green FastMix (Quanta Bioscience) and QuantStudio 3 Real-Time PCR System (Thermo Fisher Scientific). All primer sequences used are listed in Table S5. Data analysis was performed using the $2^{-\Delta\Delta Ct}$ or $2^{-\Delta Ct}$ method.

### Western blots

Mast cells were lysed in RIPA buffer (10 mM Tris–HCl, pH 8.0, 1 mM EDTA, 1% Triton X-100, 0.1% sodium deoxycholate, 0.1% SDS, 140 mM NaCl) supplemented with protease inhibitor cocktail (Sigma-Aldrich) for protein detection. Protein concentration was quantified using Pierce BCA Protein Assay Kit (Thermo Fisher Scientific).

Samples (50–100 μg) were run on SDS–polyacrylamide gels and blotted on a PVDF membrane using a wet transfer system. Membranes were blocked in 5% milk in TBST (5 mM Tris, pH 7.3, 150 mM NaCl, 0.1% Tween-20) for 30 min at RT, then incubated with primary antibodies overnight at 4°C, followed by the corresponding HRP-conjugated secondary antibody for 1 h at RT. All antibodies used are listed in Table S5. Chemiluminescence detection was performed using Clarity Western ECL Substrate (Bio-Rad) and Fusion FX7 EDGE Imaging System (WITec). Protein band intensity was quantified using ImageJ software version 1.53 h (Schindelin et al, 2012).

## Surface marker and intracellular protein stainings

For surface marker staining, mast cells were stained using CD117 (c-Kit)-APC or APC/Cy7 and FcεRIα-PE (all from BioLegend) at 1:200 dilution for 20 min on ice. For all intracellular staining experiments, cells were washed with ice-cold PBS and stained with LIVE/DEAD Fixable Aqua or Blue Dead Cell Stain (Thermo Fisher Scientific) for 20 min at RT before cell fixation. For intracellular cytokine staining, cells were stimulated with different stimuli for 4 h. Brefeldin A (Sigma-Aldrich) was added in the last 2 h of stimulation. Cells were then fixed in 4% PFA for 10 min at RT, followed by permeabilization with 1% BSA/0.5% saponin solution in PBS, and staining with the following antibodies for 20 min on ice: TNF-α-PE/Cy7, IL-6-PE or APC (both from BioLegend) and IL-13-PE or eFluor 450 (eBioscience) for BMMCs, and human TNF-α-PE (BioLegend) for HMC-1.1 and HMC-1.2 cells. For intracellular staining of HA-tagged proteins, cells were fixed and permeabilized using eBioscience Foxp3/Transcription Factor Staining Buffer Set (Thermo Fisher Scientific) according to the manufacturer's instructions. Cells were then stained with an anti-HA.11 epitope tag (BioLegend) at 1:200 dilution for 1 h at RT, followed by anti-mouse IgG (H+L)-Alexa Fluor 647–conjugated secondary antibody at 1:500 dilution for 30 min at RT. All antibodies used are listed in Table S5. Flow cytometry data were collected using FACSymphony A5 or LSRFortessa (BD Biosciences) and analyzed by FlowJo v10.6.0 (BD Biosciences).

## siRNA transfection

BMMCs were transfected with siRNA cocktails targeting Zc3h12a and/or Zc3h12c (Horizon Discovery/Dharmacon) using 100 μl Neon Transfection System Kit (Thermo Fisher Scientific) according to the manufacturer's protocol. siGLO Green Transfection Indicator (Horizon Discovery/Dharmacon) was used as a control. Briefly, cells were washed with PBS and resuspended in 100 μl of buffer R. siRNAs (200 pmol) were then added to the cell suspension, and cells were electroporated with one pulse at 1,600 V and 30 ms of width. Transfected cells were kept in antibiotic-free medium for 24 h. Downstream experiments were performed after 48–72 h. All siRNA sequences used are listed in Table S5.

## NanoString profiling

BMMCs transfected with siRNAs against Zc3h12a and/or Zc3h12c or siGLO control were stimulated with 1 μg/ml IgE/anti-DNP antibody and 0.2 μg/ml HSA-DNP antigen for 2 h, then lysed in TRI Reagent RT. Total RNA was extracted using Direct-zol RNA Microprep Kit and

quantified using Qubit RNA High Sensitivity Assay Kit and Qubit 3 Fluorometer (Thermo Fisher Scientific). Purified RNA (50 ng) was hybridized to the nCounter Myeloid Innate Immunity Panel v2 codeset (NanoString Technologies) for 16 h at 65°C. After hybridization, 30–35 μl of sample was added to the nCounter cartridge and analyzed using nCounter SPRINT Profiler (NanoString Technologies) according to the manufacturer's instructions. Data analysis was carried out using nSolver Analysis Software v4.0 (NanoString Technologies) and the Omics Playground web-based platform (Akhmedov et al, 2020). Genes with $\log_2$ fold change >0.5 and <−0.5 and $P \leq 0.05$ were considered as differentially expressed.

## CRISPR/Cas9 gene editing

BMMCs were transfected with Cas9-gRNA RNPs targeting Zc3h12a or Zc3h12c using 10 μl Neon Transfection System Kit (Thermo Fisher Scientific) according to the manufacturer's protocol, and as previously described (Leoni et al, 2023). To generate gRNAs, equal amounts (400 pmol) of crRNA and tracrRNA were mixed with nuclease-free duplex buffer and annealed by boiling at 95°C, then cooling down to RT. Three different crRNAs were selected for Zc3h12a and Zc3h12c. A non-targeting crRNA (scrambled) was used as a control (all from Integrated DNA Technologies). Cas9-gRNA RNPs were prepared by incubating either 0.5 μl of each gRNA with 1.5 μl TrueCut Cas9 Protein v2 (5 μg/μl; Thermo Fisher Scientific) or 1 μl of each gRNA with 1.5 μl recombinant Cas9-NLS (5 μg/μl, in-house) for 20 min at RT. To increase transfection efficiency, Alt-R Electroporation Enhancer (Integrated DNA Technologies) was added to the transfection mix. Cells were resuspended in 10 μl of buffer R, and electroporated with the RNPs with one pulse at 1,600 V and 30 ms of width for BMMCs, and one pulse at 1,700 V and 20 ms of width for HMC-1.1 and HMC-1.2 cells. Transfected cells were kept in antibiotic-free medium for 24 h. Downstream experiments were performed after 1 wk. All oligonucleotides used are listed in Table S5.

## Expression plasmids

Regnase-1 and Regnase-3 expression plasmids were generated using standard cloning techniques. The murine Zc3h12a coding sequence was amplified from the pRetro-Xtight-Myc-GFP-Regnase-1 plasmid (Behrens et al, 2021), whereas the Zc3h12c coding sequence was amplified from cDNA obtained from BMMCs. These were either cloned into the pCDNA3 vector (Thermo Fisher Scientific) and pCDH-EF1α-T2A-copGFP (System Biosciences) for expression in HEK293T or tagged with FLAG-HA, then cloned into the pUC57-mini vector (synthesized by GenScript) to be used as a template for in vitro mRNA transcription. The human ZC3H12A coding sequence was amplified from the pEXPR-IBA105-MCPIP1 plasmid (Behrens et al, 2018), then cloned into the pUC57-mini vector. To generate the RNase-inactive mutants, residues D141 of Regnase-1 (either human or mouse) and D252 of Regnase-3 were mutated to asparagine using QuikChange II XL Site-Directed Mutagenesis Kit (Agilent) following the manufacturer's protocol. To generate C-terminal truncations of Regnase-3, a stop codon was incorporated at the indicated sites using QuikChange II XL Site-Directed Mutagenesis Kit (Agilent) following the manufacturer's protocol. To generate the luciferase

reporter plasmid, the full-length 3'UTR of *Zc3h12a* was amplified from genomic DNA obtained from BMMCs and cloned downstream of the luciferase reporter gene in pmirGLO Dual-Luciferase miRNA Target Expression Vector (Promega).

### In vitro mRNA transcription

To express Regnase-1 and -3, pUC57-mini plasmids harboring the WT or mutated coding sequence downstream of the T7 promoter were first transcribed into mRNA using HiScribe T7 ARCA mRNA Kit (New England BioLabs) according to the manufacturer's protocol. Briefly, the plasmids were linearized by digestion with SpeI (New England BioLabs), followed by purification using NucleoSpin Gel and PCR Clean-up Kit (Macherey-Nagel). Linearized DNA templates (1 $\mu$g) were then assembled together with ARCA/NTP mix, T7 RNA polymerase mix (both included in the kit), and pseudo-UTP (Jena Bioscience) for the IVT reaction. mRNA synthesis was completed by removing the DNA template through DNase I treatment, followed by poly(A) tailing by adding poly(A) polymerase and buffer (all included in the kit) to the reaction mix. All reactions were each done at 37°C for 30 min. Resulting mRNA products were purified using Monarch RNA Cleanup Kit (New England BioLabs), visualized on 1% TBE gel, and quantified using NanoDrop 2000. BMMCs were transfected with Regnase-1 or Regnase-3 (WT or RNase-inactive mutant D141N or D252N) IVT mRNA using 10 $\mu$l Neon Transfection System Kit (Thermo Fisher Scientific) according to the manufacturer's protocol. IVT mRNA from the pUC57-mini plasmid expressing ZsGreen alone was used as a control. Briefly, cells were washed with PBS and resuspended in 10 $\mu$l of buffer R. IVT mRNA (3 pmol for BMMCs and 0.25 pmol for HMC-1.1 and HMC-1.2 cells) and 10U RNase inhibitor (Promega) were then added to the cell suspension. Cell electroporation was performed with one pulse at 1,600 V and 30 ms of width for BMMCs, and one pulse at 1,700 V and 20 ms of width for HMC-1.1 and HMC-1.2 cells. Transfected cells were kept in antibiotic-free medium, and downstream experiments were performed within 24 h.

### Luciferase reporter assays

HEK293T cells were transfected with 3 $\mu$g of a pmirGLO plasmid containing the 3'UTR of *Zc3h12a* or *Tnf* (plasmid 207127; Addgene) (Leoni et al, 2023) and 1 $\mu$g of pCDNA3-Regnase-1 WT/D141N or pCDH-Regnase-3 WT/D252N or the corresponding empty control plasmids using polyethylenimine (PEI). After 48 h, luciferase activity was analyzed using Dual-Luciferase Reporter Assay System (Promega) according to the manufacturer's protocol and measured using GloMax Discover (Promega).

### mRNA stability assay

BMMCs were treated with 1 $\mu$g/ml IgE/anti-DNP (Sigma-Aldrich) for 30 min followed by stimulation with 0.2 $\mu$g/ml HSA-DNP (Sigma-Aldrich) for 30 min. Transcription was then blocked by incubating the cells with 10 $\mu$g/ml actinomycin D (Sigma-Aldrich). Cells were harvested at different time points (15, 30, and 60 min), and RNA isolation and RT–qPCR were performed as mentioned above. mRNA expression was calculated using the $2^{-\Delta Ct}$ and $2^{-\Delta\Delta Ct}$ method,

normalizing against *Gapdh* expression. When possible, the mRNA decay rate (half-life, $t^{1/2}$) was calculated by non-linear regression curve fitting (one phase decay) using GraphPad Prism version 9 (GraphPad).

### Cell viability and apoptosis assays

Cell viability was measured by staining cells with LIVE/DEAD Fixable Aqua Dead Cell Stain or Blue Dead Cell Stain (Thermo Fisher Scientific) for 20 min at RT. Cell apoptosis was measured by co-staining cells with Annexin V and 7-AAD using PE Annexin V Apoptosis Detection Kit I (BD Biosciences), according to the manufacturer's protocol. All experiments were performed without acute stimulation.

### Cell proliferation and cell cycle analysis

Cell proliferation was measured by BrdU incorporation for 12–16 h at 37°C, followed by BrdU staining using APC BrdU Flow Kit (BD Biosciences), according to the manufacturer's protocol. Cell cycle progression was analyzed by co-staining cells with 7-AAD. All experiments were performed without acute stimulation.

### Cell degranulation assay

BMMCs were treated with 1 $\mu$g/ml IgE/anti-DNP (Sigma-Aldrich) and 0.2 $\mu$g/ml HSA-DNP (Sigma-Aldrich) for 30 min. Cell degranulation was measured by Annexin V staining using PE Annexin V Apoptosis Detection Kit I (BD Biosciences), according to the manufacturer's protocol.

### RNA sequencing

BMMCs were transfected with *Zc3h12a* RNPs or scrambled RNPs as a control (four independent biological replicates). After 1 wk in culture, cells were lysed in TRI Reagent RT (Molecular Research Center) and RNA was extracted using Direct-zol RNA Microprep Kit (Zymo Research). After poly(A) mRNA enrichment and Tecan Revelo mRNA Library Preparation, mRNA sequencing using Illumina NovaSeq 6000 (2 × 50-bp reads) was outsourced to Next Generation Sequencing Platform at the University of Bern (Switzerland). Read quality control was assessed using fastqc v.0.11.9 (http://www.bioinformatics.babraham.ac.uk/projects/fastqc/) and RSeQC v.4.0.0 (Wang et al, 2012), followed by read mapping to the reference genome Mus_musculus.GRCm39.107 using HiSat2 v.2.2.1 (Kim et al, 2015). Counts were generated and corrected for batch effects using featureCounts v.2.0.1 (Liao et al, 2014) and removeBatchEffect function in R (Ritchie et al, 2015), respectively. Differential expression was analyzed by combining three DE methods: DESeq2 (Wald), edgeR (QLF), and limma (trend), using the Omics Playground platform (Akhmedov et al, 2020). Genes with $\log_2$ fold change >0.5 and <−0.5 and $P \leq 0.05$ were considered differentially expressed and included in the gene ontology (GO) analysis performed using DAVID Bioinformatics Resource v2023q2 (Huang da et al, 2009; Sherman et al, 2022). The top 10 GO enriched terms in up-regulated and down-regulated genes were graphically represented (for up-

regulated genes, only categories with >10 genes were considered). Data visualization was performed with RStudio version 4.1.

## Meta-analysis of RBP expression

Re-analysis of RNA-seq data from Li et al (2021) was performed using the DESeq2 package in R (Love et al, 2014). Genes encoding for RBPs identified in more than one RBPome listed in the RBP2GO database (Caudron-Herger et al, 2021) and with the sum of counts in all samples ≥10 were included in the differential expression analysis. RBP genes with $\log_2$ fold change >0.5 and <−0.5 and $P \leq 0.05$ were considered differentially expressed and included in functional categorization analysis using PANTHER 17.0 (Mi et al, 2019). First, differentially expressed RBP genes broadly categorized under the GO term RNA metabolic process (GO:0016070) were identified. This was followed by more specific categorization using the following GO terms: regulation of RNA stability (GO:0043487), RNA modification (GO:0009451), RNA processing (GO:0006396), RNA localization (GO:0006403), and translation (GO:0006412). Data visualization was performed with RStudio version 4.1. ATAC-seq data from Li et al (2021) were visualized using Integrative Genomics Viewer (IGV) 2.16.0 (Robinson et al, 2011).

## Lentiviral preparation and cell transduction

To generate lentivirus, HEK293T cells were transfected with lentiviral vectors together with the packaging vectors psPAX2 and pMD2.G (plasmids 12260 and 12259 by Didier Trono; Addgene) using PEI. After 48 h, lentiviral particles in the supernatant were collected and concentrated using a PEG-8000 solution as previously described (Lo & Yee, 2007). For co-expression experiments, BMMCs were first transduced with pScalps-Regnase-1 D141N lentivirus and treated with puromycin for 48 h, followed by a second transduction with pScalps-Regnase-3 D252N lentivirus. For Cre recombinase expression, the pScalps-EGFP-Cre recombinase vector (Leoni et al, 2023) (plasmid 207132; Addgene) was used to transduce WT and *Zc3h12a*<sup>fl/fl</sup> BMMCs, for at least 96 h. Flow cytometry–based analyses were done by gating on EGFP⁺ cells, whereas Western blot experiments were performed on EGFP⁺ cell population sorted using BD FACSymphony S6 Cell Sorter (BD Biosciences).

## Immunoprecipitation and enzymatic digestion

BMMCs ($2 \times 10^7$) were activated with 2 nM PMA and 200 nM ionomycin for 1 h. Cells were washed with ice-cold PBS and lysed in IP buffer (50 mM Tris, pH 7.5, 150 mM NaCl, 1 mM EDTA, 0.25% NP-40, 1X protease inhibitor [Sigma-Aldrich], 1X phosphatase inhibitor [Sigma-Aldrich]). Lysates were cleared of cell debris by centrifugation at 10,000*g* for 10 min at 4°C, followed by pre-clearing with 20 µl Dynabeads Protein G (30 mg/ml; Invitrogen) at 4°C with rotation for 2 h. 1 mg of pre-cleared lysates was incubated with 50 µl Dynabeads Protein G (30 mg/ml) and 5 µg Regnase-1 (R&D Systems) or Regnase-3 (Helmholtz Munich Monoclonal Antibody Core Facility) antibody at 4°C with rotation for 16 h. Mouse IgG (Sigma-Aldrich) or rat IgG (Thermo Fisher Scientific) antibodies were used as isotype controls, respectively. Immunoprecipitated proteins were pulled down using DynaMag-2 Magnet (Thermo Fisher Scientific) and washed two

times with IP buffer with 0.1% NP-40, followed by two washes with detergent-free wash buffer (50 mM Tris, pH 7.5, 150 mM NaCl, 1X protease inhibitor, 1X phosphatase inhibitor).

On-bead digestion of immunoprecipitated proteins was performed as follows. Beads were resuspended in 8 M urea, 50 mM ammonium bicarbonate buffer. Proteins were then reduced with 10 mM dithiothreitol for 60 min at 37°C and alkylated with 50 mM iodoacetamide for 30 min at RT. Digestion was carried out in 8 M urea, 50 mM ammonium bicarbonate buffer for 2 h at 37°C with 1 µg of Lys-C (FUJIFILM Wako Chemicals), after which the digestion buffer was diluted to final 2 M urea with 50 mM ammonium bicarbonate. 1 µg of trypsin (Promega) was added for overnight digestion at 37°C. All steps were performed in agitation to avoid bead precipitation. Digestion was stopped by adding acetonitrile to 2% and trifluoroacetic acid to 0.3%, and the beads were collected with DynaMag-2 Magnet (Thermo Fisher Scientific). Digested peptides were purified by loading the supernatant into C18 StageTips (Rappsilber et al, 2007), and eluted with 80% acetonitrile, 0.5% acetic acid. Finally, the elution buffer was evaporated by vacuum centrifugation and purified peptides were resolved in 2% acetonitrile, 0.5% acetic acid, 0.1% trifluoroacetic acid for single-shot LC-MS/MS measurements.

## LC-MS/MS

Peptides were separated on an EASY-nLC 1200 HPLC system (Thermo Fisher Scientific) coupled online via a nanoelectrospray source (Thermo Fisher Scientific) to a Q Exactive HF mass spectrometer (Thermo Fisher Scientific). Peptides were loaded in buffer A (0.1% formic acid) into a 75 µm inner diameter, 50 cm length column, packed in-house with ReproSil-Pur 120 C18-AQ 1.9 µm resin (Dr. Maisch HPLC GmbH), and eluted over a 150-min linear gradient of 5–30% buffer B (80% acetonitrile, 0.1% formic acid) at a flow rate of 250 nl/min. Xcalibur software (Thermo Fisher Scientific) operated the Q Exactive HF in a data-dependent mode with a survey scan range of 300–1,650 m/z, resolution of 60,000 at 200 m/z, maximum injection time of 20 ms, and AGC target of $3 \times 10^6$. The top 10 most abundant ions with charge 2–5 were isolated with a 1.8 m/z isolation window and fragmented by higher energy collisional dissociation (HCD) at a normalized collision energy of 27. MS/MS spectra were acquired with a resolution of 15,000 at 200 m/z, maximum injection time of 55 ms, and AGC target of $1 \times 10^5$. Dynamic exclusion was set to 30 s to avoid repeated sequencing.

## LC-MS/MS data analysis

MS raw files were processed using MaxQuant software v.1.6.7.0 (Cox & Mann, 2008). Peptides and proteins were identified with a 0.01 false discovery rate by employing the integrated Andromeda search engine (Cox et al, 2011) to search spectra against the mouse UniProt database (July 2019) and a common contaminants database (247 entries). Enzyme specificity was set as "Trypsin/P" with a maximum of two missed cleavages and minimum length of seven amino acids. N-terminal protein acetylation and methionine oxidation were set as variable modifications, and cysteine carbamidomethylation as a fixed modification. Match between runs was enabled to transfer

identifications across samples based on mass and normalized retention times, with a matching time window of 0.7 min and an alignment time window of 20 min. Label-free protein quantification (LFQ) was performed with the MaxLFQ algorithm (Cox et al, 2014) with a minimum required peptide ratio count of 1. Data analysis was performed using Perseus software v.1.6.2.3 (Tyanova et al, 2016). Data were pre-processed by removing proteins only identified by site, reverse hits, and potential contaminants. After $\log_2$ transformation of LFQ intensities, biological replicates of each experimental condition were grouped and proteins were filtered for a minimum of three valid values in at least one group. Missing data points were then replaced by imputation from a normal distribution with 0.3 width and 1.8 downshift, and a two-sided two-samples $t$ test (0.05 false discovery rate, 250 randomizations) was used to identify significant changes in protein intensity between each immunoprecipitation experiment and its corresponding isotype control.

### Regnase-1 and -3 co-transfection in HEK293T cells

HEK293T cells were co-transfected with pCDNA3-Regnase-1 and pCDNA3-Regnase-3 plasmids (10 $\mu$g each, full-length or C-term truncated) using polyethylenimine (PEI). After 48 h, cells were harvested for immunoprecipitation experiments using 5 $\mu$g anti-Regnase-1 (R&D Systems), anti-Regnase-3 (Helmholtz Munich Monoclonal Antibody Core Facility), anti-HA (BioLegend), or anti-FLAG (Sigma-Aldrich) antibodies. For experiments with RNase treatment, pre-cleared lysates were treated with 100 $\mu$g/ml RNase A (Zymo Research) and RNase I (Thermo Fisher Scientific) at 37°C for 30 min before incubation with the antibody–bead complex.

### Immunofluorescence

Cells were fixed with 4% PFA in PBS for 10 min, then cytospun on a coverslip. Cells were permeabilized with 0.2% Triton X-100 in PBS for 15 min and blocked with 5% BSA in PBS for 30 min. Cells were then stained with anti-Regnase-1 (R&D Systems) or anti-Regnase-3 (Helmholtz Munich Monoclonal Antibody Core Facility) at 1:100 dilution for 1.5 h, followed by anti-mouse IgG (H+L)-Alexa Fluor 568– or anti-rat IgG (H+L)-Alexa Fluor 647–conjugated secondary antibody (Thermo Fisher Scientific) at 1:500 dilution for 30 min. All incubations were done at RT. Stained cells were washed three times with PBS and once with water, followed by nucleus counterstaining and mounting on microscopy slides using Vectashield (Vector Laboratories) with DAPI. All antibodies used are listed in Table S5. Confocal microscopy imaging was performed using Leica Stellaris SP8 confocal laser scanning microscope with Leica HCX PL APO 40 × 1.30 oil or 63 × 1.40 oil objectives (Leica Microsystems). Image acquisition and processing were done using Leica Application Suite X (Leica Microsystems) and ImageJ version 1.53 h (Schindelin et al, 2012) software.

### ELISA

1 × $10^5$ cells were stimulated with 1 $\mu$g/ml IgE/anti-DNP and 0.2 $\mu$g/ml HSA-DNP for 6 h in a 96-well plate before collection of the supernatant and measurement of TNF release using Mouse TNF-$\alpha$ High Sensitivity ELISA (eBioscience), following the manufacturer's instructions.

### Data analysis

All data shown consist of biological (not technical) replicates. Data representation and statistical analysis were performed using GraphPad Prism v9. All source data for all figures are provided in the **Source Data Files**.

## Data Availability

All raw data are either contained within this article or deposited in Gene Expression Omnibus (GEO) with accession number GSE240210 and in ProteomeXchange via the PRIDE database (Perez-Riverol et al, 2022) with dataset identifier PXD051849.

## Supplementary Information

## Acknowledgements

Work on RNA-binding proteins in the SM laboratory was funded by the Swiss National Science Foundation grant number 310030L_189352 and the Aldo and Cele Daccò Foundation (to S Monticelli). The authors would like to thank Emina Džafo for plasmid design, as well as Pamela Nicholson (Next Generation Sequencing Platform, University of Bern) for RNA-seq.

### Author Contributions

M Bataclan: conceptualization, data curation, formal analysis, investigation, visualization, and writing—original draft, review, and editing.
C Leoni: conceptualization, formal analysis, investigation, and visualization.
SG Moro: data curation, formal analysis, and visualization.
M Pecoraro: formal analysis and investigation.
EH Wong: resources, formal analysis, validation, and methodology.
V Heissmeyer: resources, funding acquisition, validation, methodology, and writing—review and editing.
S Monticelli: conceptualization, formal analysis, supervision, funding acquisition, visualization, project administration, and writing—original draft, review, and editing.

### Conflict of Interest Statement

The authors declare that they have no conflict of interest.

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
