## [Reviewer comments · Life Science Alliance]

Life Science Alliance

Crosstalk between Regnase-1 and -3 shapes mast cell survival and cytokine expression

Marian Bataclan, Cristina Leoni, Simone Moro, Matteo Pecoraro, Elaine Wong, Vigo Heissmeyer, and Silvia Monticelli
DOI: <https://doi.org/10.26508/lsa.202402784>

Corresponding author(s): *Silvia Monticelli, Institute for Research in Biomedicine*

Review Timeline:

Submission Date:	2024-04-22
Editorial Decision:	2024-05-13
Revision Received:	2024-05-17
Accepted:	2024-05-22

Scientific Editor: *Eric Sawey, PhD*

Transaction Report:

Please note that the manuscript was reviewed at *Review Commons* and these reports were taken into account in the decision-making process at *Life Science Alliance*.

Review
COMMONS

Reviews

Review #1

1. Evidence, reproducibility and clarity:

****Summary:****

Mast cells play a role in exerting effector functions in the immune response. However, the functioning of RNA-binding proteins (RBPs) in the mast cells is not well understood. The authors focused on the Regnase family of RNA-degrading enzymes and worked towards unraveling their functions. Upon activating mast cells, the authors observed a significant induction of Regnase-3 among RBPs. Furthermore, they found that Regnase-3 directly controls the well-known Regnase-1, revealing its role in regulating homeostasis and inflammatory responses in mast cells.

****Major comments:****

This experiment has been meticulously designed, and the reliability of the presented data is quite high. While the observations are specific to mast cells, the insights gained could provide valuable information about the interrelation within the Regnase family across other immune cell types. However, there are three main concerns raised by the reviewer:

The first concern revolves around the use of the IVT system for the expression of Regnase-1 and Regnase-3. Is there evidence confirming the expression of Regnase-3 as a full-length protein, as shown in Figures 3g and 3h? The APC-HA signal by FACS could be positive even for degradation products alone. Assuming Regnase-3 is not expressed in its full-length, it might lead to results similar to the control. Given the larger molecular weight of Regnase-3 compared to Regnase-1, it is crucial to demonstrate sufficient expression through IVT. In particular, the Western blot data for Regnase-3 in some cases confirm the full length of the product, while in other cases more degradation products appear.

The second concern arises from the co-immunoprecipitation experiment in Figure 4i. While the experiment detects Regnase-1 co-precipitating with Regnase-3, the reverse-precipitating Regnase-1 with Regnase-3 shows a signal comparable to IgG, indicating background noise. Further improvement is needed in this aspect. Is it possible that your antibody against Regnase-1 also binds to Regnase-3?

The third concern is related to the evaluation of Regnase-1 degranulation in Figure 5g, where there appears to be some variability. Including Regnase-3 and conducting mutual evaluations could enhance the reliability of the results, possibly addressing this variability.

****Minor comments:****

Figure 6e-g also seems to vary, and the effects on cell death and cell proliferation tend to be somewhat milder.

Certainly, their IF images (Figure S2 and Figure S5) suggests that Regnase-1 and Regnase-3 have no or only a weak interaction.

Is the input to the immunoprecipitation from whole cell lysate or is it cleared with a low-speed (or high-speed) centrifugation?

2. Significance:

The following aspects are important: Understanding the regulatory mechanisms mediated by RNA-binding proteins (RBPs) has gained significant attention in recent years. While it is inferred that RBPs control specific RNAs, many uncertainties remain about how they function in various RNA metabolism processes. The Regnase family is known to operate within the mechanism of RNA stability. While lower organisms have only one type, mammals have evolved to have four types. Understanding the implications of this family expansion is highly valuable, shedding light on how the RNA within our body's cells is regulated.

- Advance: In this study, a strength lies in the meticulous examination of the relationship between Regnase-1 and Regnase-3 by handling them similarly in the context of mast cells. However, because of the multifaceted effects of

Regnase-1 on cell proliferation, cell death, and the cell cycle, significant progress in understanding it has yet to be made. Going forward, the replication of immune responses in mast cells at the animal level holds the potential to further deepen our comprehension of RBPs through the Regnase family.

This study complements two previous investigations on Regnase-3 (PMID: 34215755, 31126966) while specifically focusing on mast cells. The findings align with the prevailing perspective that emphasizes the significance of Regnase-1 among the Regnase family.

- Audience: Basic research

- Immunology, Molecular Biology, Genetic Engineering

Review #2

1. Evidence, reproducibility and clarity:

****Summary:****

Bataclan et al. studied the role of Regnase-1 and -3 in the mast cell (MC) function in vitro. Using siRNA, CRISPR-mediated depletion, and overexpression models in, mainly, IL-3-cultured bone-marrow-derived mast cells, they beautifully demonstrated that Regnase-1 and, to a lesser extent, Regnase-3 suppressed the expression of TNF in MCs in response to IgE-stimulation. Regnase-1 also had roles in MC survival and IgE-stimulation-induced degranulation. Regnase-3 seems to have mixed effects on MC functions as the protein's primary targets include both TNF and Regnase-1 mRNAs.

****Major comments:****

1. The authors mainly used BMMCs generated by four weeks of cultivation of bone marrow cells with IL-3; at this point, MCs are considered "less mature". Could the authors reproduce the primary data (the role of Regnase-1 on TNF, cell survival, and degranulation) even if they used more mature MCs (e.g., extended cultivation with IL-3 (6-7 weeks))?
2. TNF mRNA is not considered a primary target for Regnase-1 in other cell types, such as macrophages or fibroblasts. Although this reviewer does not doubt that the TNF expression level is controlled by Regnase-1 in MCs, more evidence is needed to conclude that TNF is a "primary target" for Regnases. They can determine the stability of "endogenous" TNF mRNA in Regnase-depleted cells, as in Figure 4d.
3. The role of Regnase-1 on the MC degranulation is less pronounced. Could the authors show the same result using another assay, such as a FACS-based assay excluding dead or dying cells?
4. Have the authors had a chance to look into which protease(s) cleaved Regnase-1 in IgE-stimulated MCs?

2. Significance:

General assessment:

This study investigated the role of Regnase-1 and -3 in the mast cell (MC) function in vitro. The strength of the study is that they used several methods to manipulate the expression levels of Regnases in the cells (siRNA, CRISPR, and Regnase-1 overexpression) and obtained consistent results. Although the study is well designed and the results are beautiful, the BMMCs the authors used are less mature MCs and do not represent the cells in the mammalian body well. Also, this study only focused on in vitro mouse-derived MCs, and human MCs or in vivo roles of MC Regnases were not studied.

Advance:

The role of Regnase-1 in several cell types has already been shown, including the populations involved in type-2 immunity (Th2 and ILC2). However, this study might be the first to investigate the role of Regnases in MCs. The roles of Regnases in MCs (control of mRNA stability through the RNase activity) presented are in line with the previous studies.

Audience:

The primary audience of this study may be basic researchers studying type-2 immunity or MC biology. Because MCs are an essential cell population in several allergic disorders, some clinicians who care for allergic patients might also be interested in this study. However, main audiences may be relatively limited to specific fields like immunology and allergology.

This reviewer's main field of expertise is basic research in immunology and allergology. More specific keywords are MCs, ILC2, IgE, type-2 cytokines, and mouse models.

Review #3

1. Evidence, reproducibility and clarity:

****Summary:****

Bataclan et al. provide extensive in vitro work on Regnase1/3 function in mast cells. They make use of in vitro differentiated mast cells from bone marrow (BMMC) and in vitro cultures from primary peritoneal mast cells from mice. They show robustly Reg1/3 upregulation by transcript and protein upon activation stimulation in these cultured mast cells. Knockdown and CRISPR editing of Reg1/3 show enhanced inflammatory signaling. TNF is identified as a direct target of Reg1/3 nuclease activity. Reg3 is implicated as a negative regulator of Reg1. Overall, this work highlights Reg1/3 as novel players in mast cell biology that control inflammatory signaling including TNF.

****Major comments:****

- The result section requires more consistent & sufficient information on the assays (in vitro, ex vivo, species, etc.) particularly for the meta-analysis in the beginning. Relatedly, can the authors assume cross-species conservation of Reg1/3 in mast cells across mammals to make general conclusions on the functions or should this be limited mostly to mouse observations? It would be helpful to mention the species studied in the abstract.
- Reg1 protein levels appear to be more stable than Reg3 protein levels (mirroring the transcript) Fig1a+c. Is the half-life of Reg3 shorter than Reg1? What is functional impact of Reg1 regulation at the transcriptional level considering the protein half-lives?
- MFI comparisons can only be used on unimodal populations (not bimodal i.e. Fig3b). Also, contour flow plots should show outliers.
- Is TNF functionally secreted from WT vs KD/KO mast cells under these in vitro conditions? This is not shown and seems quite an important aspect given the focus on TNF regulation (other secreted factors could also be considered). Degranulation appears to be lower in Fig5g. Does Reg1 overexpression reduces TNF expression and secretion?
- Catalytic variants of Reg1/3 appear to be only tested in WT cells. Wouldn't it be worthwhile to test these in KO cells? The absolute protein levels of ectopic and endogenous are not entirely clear and only shown for WT Reg3 in Fig4b.
- Fig2b shows upregulation of Reg1 transcript is not enhanced upon Reg3 knockdown, but it is enhanced in Fig4a? How are these assays different?
- The authors implicate Reg3 to regulate Reg1 by transcript. What is the role of the protein interaction between Reg1 and 3? Why is the protein interaction not seen in the IF experiments? Would other cell systems (maybe mouse) be more suitable.
- Cell death increases after Reg1/3 knockdowns; can this be rescued with Reg1/3 reinstatement? Is the protein level of Reg1 important for this; can this be dosed? Are the assays for cell death/proliferation done in comparison to unstimulated resting cells, cultured conditions and IgE stimulated mast cells? Minor point: A single assay of cell death is sufficient in the main figures; same for proliferation assays.
- It is not entirely clear how the RNA-seq data with CRISPR KO in Fig6 is different than the nanostring data with Knockdowns shown earlier in Fig2? Were unstimulated cells not used as a control before? A floxed mouse model for Reg1 is introduced at the end, but would have been useful for many assays. OPTIONAL: are in vivo experiments of interest to confirm the findings? The conclusions that Reg1/3 regulates physiological mast cell responses might be a reach otherwise.

****Minor comments:****

- Some of the Knockdown and CRISPR validation main figures are redundant and might be better suited for the supplement.
- The MW in the western blot requires lines to indicate the exact location of the ladder.
- Fig2 title refers as loss of Reg1/3 using siRNA. The term is typically used in the context of genetic ablation and not for knockdowns.
- Gene locus figures of Reg1/3 are confusing why are only E3 and E6 shown and not all Exons (Fig2a et al)?
- Are biological replicates or technical replicates used in Fig1?
- Is a student's t test the appropriate statistical analysis for most of the analyses?
- An explanation why nanostring and RNA-seq are both used would be helpful.

2. Significance:

Bataclan et al investigate Reg1/3 function in mast cells, that are of interest to elucidate the regulation of mast cell effector functions. Reg1/3 were identified as novel regulators of murine mast cells. Independent and complementary use of Knockdowns and CRISPR deletions provide robust data. Also, the use of two independent mast cell populations and different stimuli is of interest, albeit they are not used in every assay. Altogether, the in vitro data are robust and rigorous. On the flipside, in vivo data is not provided. The translational impact would be higher if findings are tested in in vivo models including preclinical applications (as discussed in text). Mouse studies are often indicative of mechanisms in higher mammals, but it would help to lay this out more clearly. Therefore, the connection to humans is less clear. The type of work presented here would be best described as basic research. This review has been assessed with the following expertise: mouse/human immunology, mouse models, immune cell signaling, immune effector functions, flow cytometry.

Referee #1

This experiment has been meticulously designed, and the reliability of the presented data is quite high. While the observations are specific to mast cells, the insights gained could provide valuable information about the interrelation within the Regnase family across other immune cell types.

In this study, a strength lies in the meticulous examination of the relationship between Regnase-1 and Regnase-3 by handling them similarly in the context of mast cells. However, because of the multifaceted effects of Regnase-1 on cell proliferation, cell death, and the cell cycle, significant progress in understanding it has yet to be made. Going forward, the replication of immune responses in mast cells at the animal level holds the potential to further deepen our comprehension of RBPs through the Regnase family.

This study complements two previous investigations on Regnase-3 (PMID: 34215755, 31126966) while specifically focusing on mast cells. The findings align with the prevailing perspective that emphasizes the significance of Regnase-1 among the Regnase family.

- Audience: Basic research, Immunology, Molecular Biology, Genetic Engineering
-

Referee #2

The strength of the study is that they used several methods to manipulate the expression levels of Regnases in the cells (siRNA, CRISPR, and Regnase-1 overexpression) and obtained consistent results. Although the study is well designed and the results are beautiful, the BMDCs the authors used are less mature MCs and do not represent the cells in the mammalian body well. Also, this study only focused on in vitro mouse-derived MCs, and human MCs or in vivo roles of MC Regnases were not studied.

The role of Regnase-1 in several cell types has already been shown, including the populations involved in type-2 immunity (Th2 and ILC2). However, this study might be the first to investigate the role of Regnases in MCs. The roles of Regnases in MCs (control of mRNA stability through the RNase activity) presented are in line with the previous studies.

The primary audience of this study may be basic researchers studying type-2 immunity or MC biology. Because MCs are an essential cell population in several allergic disorders, some clinicians who care for allergic patients might also be interested in this study. However, main audiences may be relatively limited to specific fields like immunology and allergology.

Referee #3

Reg1/3 were identified as novel regulators of murine mast cells. Independent and complementary use of Knockdowns and CRISPR deletions provide robust data. Also, the use of two independent mast cell populations and different stimuli is of interest, albeit they are not used in every assay. Altogether, the in vitro data are robust and rigorous.

On the flipside, in vivo data is not provided. The translational impact would be higher if findings are tested in in vivo models including preclinical applications (as discussed in text). Mouse studies are often indicative of mechanisms in higher mammals, but it would help to lay this out more clearly. Therefore, the connection to humans is less clear. The type of work presented here would be best described as basic research.

1. General Statements [optional]

We thank the Reviewers for providing constructive comments that allowed us to extend our study and further strengthen our conclusions. We are delighted that all three Reviewers appreciated the high quality of our data. As outlined below, we now addressed in full their remaining concerns. Furthermore, we widened the overall scope of our findings by adding mechanistic data related to the physical interaction between Regnase-1 and -3 proteins, as well as data from human mast cells. All changes are highlighted in yellow in the text. Overall, these new data further extend and strengthen our previous conclusions, and we are grateful to the Reviewers for their suggestions.

Reviewer #1 (Evidence, reproducibility and clarity (Required)):

Summary:

Mast cells play a role in exerting effector functions in the immune response. However, the functioning of RNA-binding proteins (RBPs) in the mast cells is not well understood. The authors focused on the Regnase family of RNA-degrading enzymes and worked towards unraveling their functions. Upon activating mast cells, the authors observed a significant induction of Regnase-3 among RBPs. Furthermore, they found that Regnase-3 directly controls the well-known Regnase-1, revealing its role in regulating homeostasis and inflammatory responses in mast cells.

Major comments:

This experiment has been meticulously designed, and the reliability of the presented data is quite high. While the observations are specific to mast cells, the insights gained could provide valuable information about the interrelation within the Regnase family across other immune cell types. However, there are three main concerns raised by the reviewer:

We thank the Reviewer for appreciating the high quality and reliability of our data and experimental design, and for commenting positively about the overall knowledge gain.

The first concern revolves around the use of the IVT system for the expression of Regnase-1 and Regnase-3. Is there evidence confirming the expression of Regnase-3 as a full-length protein, as shown in Figures 3g and 3h? The APC-HA signal by FACS could be positive even for degradation products alone. Assuming Regnase-3 is not expressed in its full-length, it might

lead to results similar to the control. Given the larger molecular weight of Regnase-3 compared to Regnase-1, it is crucial to demonstrate sufficient expression through IVT. In particular, the Western blot data for Regnase-3 in some cases confirm the full length of the product, while in other cases more degradation products appear.

The Reviewer raises an interesting point: the Regnase-3 signal obtained with the antibody indeed spans a range of protein sizes. These do not appear to be degradation events due to technical reasons, since we observe no degradation of other proteins in our samples, and the same complex pattern of bands was observed also by others in different cell types (von Gamm et al., 2019). All signals appear to be specific, since they are present in different experimental conditions, including in vitro transcribed mRNA (IVT)-expressed protein (**Figure 4b**), and they all disappear in Regnase-3 knock-out cells (**Figure 2c, 3d** and (von Gamm et al., 2019)). Moreover, the same pattern of bands was obtained using an anti-Regnase-3 and anti-HA antibody to detect tagged-Regnase-3 (see **Figure for Reviewer** below). Since Regnase-1 is also known to undergo proteolytic cleavage, one possibility is that multiple cleavage products of Regnase-3 are present in cells. However, whether Regnase-3 undergoes post-translational regulation remains unknown at this stage. This is mentioned in the manuscript (**page 4**).

[Figure removed by editorial staff per authors' request]

Concerning the specific point of the Reviewer relative to the expression of Regnase-3 in IVT mRNA transfected cells, we analyzed Regnase-3 expression by Western blot (**Figure 4b**), showing that indeed the pattern of bands observed for IVT mRNA-expressed Regnase-3 is comparable to that of endogenous Regnase-3. As discussed in the manuscript (**page 6**), the levels of Regnase-3 overexpression that we could achieve are somewhat lower compared to Regnase-1 (**Figure 3g**). To increase the expression of Regnase-3, we titrated the amount of IVT mRNA transfected into cells.

However, we would like to highlight that the levels of IVT mRNA-expressed Regnase-3 are overall much higher compared to the endogenous protein (**Figure for Reviewer** below),

confirming ectopic overexpression and therefore formally excluding that a lack of efficacy is simply due to low Regnase-3 expression. Rather, Regnase-3 appears to have functions that are distinct from Regnase-1.

[Figure removed by editorial staff per authors' request]

The second concern arises from the co-immunoprecipitation experiment in Figure 4i. While the experiment detects Regnase-1 co-precipitating with Regnase-3, the reverse-precipitating Regnase-1 with Regnase-3 shows a signal comparable to IgG, indicating background noise. Further improvement is needed in this aspect. Is it possible that your antibody against Regnase-1 also binds to Regnase-3?

We agree with the Reviewer that the fact that Regnase-1 is detectable upon Regnase-3 immunoprecipitation and not the opposite is interesting. First, based on the results of numerous overexpression and knock-out experiments, we found no evidence that the Regnase-1 antibody might also bind to Regnase-3. This is consistent also with the datasheet for this antibody (Bio-Techne Human/Mouse MCPIP1 antibody MAB7875) specifically stating the lack of cross-reactivity with Regnase-3 (see link and screenshot from the website below) (https://www.bio-technique.com/p/antibodies/human-mouse-mcpip1-antibody-604421_mab7875).

Product Summary for Human/Mouse MCPIP1 Antibody

Immunogen	E. coli -derived recombinant human MCPIP1 Asp426-Glu599 Accession # Q5D1E8
Specificity	Detects human MCPIP1 in direct ELISAs and human and mouse MCPIP1 in Western blots. In direct ELISAs, no cross-reactivity with recombinant human MCPIP3 is observed.
Clonality	Monoclonal
Host	Mouse
Isotype	IgG _{2A}

To experimentally address this specific point, we first performed intracellular staining of HEK293 cells transfected with Regnase-1 or -3, or both, and stained them with both antibodies simultaneously. The results showed that both antibodies are very specific and no cross-reactivity is detected. The same results were obtained upon intracellular staining of mast cells transduced to overexpress catalytically inactive Regnase-1 D141N and Regnase-3 D252N (new **Supplementary Figure 7b-c**. This is now described also in the text (**page 7**).

[Figure removed by editorial staff per authors' request]

Next, we performed additional control immunoprecipitations using an anti-FLAG-tag antibody to immunoprecipitate FLAG-Regnase-1 or FLAG-Regnase-3, followed by detection with anti-Regnase-3 or anti-Regnase-1 antibody, respectively. We found that Regnase-3 co-immunoprecipitated with FLAG-Regnase-1 when using the anti-FLAG antibody (new **Supplementary Figure 7d**, reported below). Vice versa, Regnase-1 was not detectable upon immunoprecipitation of FLAG-Regnase-3 when using the anti-FLAG antibody, fully confirming our previous mass spectrometry data. We conclude that while Regnase-1 interacts with fewer proteins, including itself (Yokogawa et al., 2016) and Regnase-3, the protein-protein interactions of Regnase-3 are more promiscuous. Of these, Regnase-1 appears not be the predominant interactor, but rather proteins involved in translation, regulation of ribosomes and 14-3-3 proteins, as discussed also in the text (**page 7**).

[Figure removed by editorial staff per authors' request]

Moreover, using mast cells co-transduced to overexpress catalytically inactive Regnase-1 D141N and Regnase-3 D252N, we confirmed that the Regnase-1 and-3 interaction occurs also in primary mast cells (new **Figure 4j**, reported below).

[Figure removed by editorial staff per authors' request]

Finally, to gain insights into the region(s) involved in protein-protein interactions between Regnase-1 and -3, we generated a panel of Regnase-3 C-terminal truncation mutants (schematic representation in **Figure 4k**). We found that all truncations were expressed at comparable levels in HEK293 cells (**Figure 4l**, reported below). Co-immunoprecipitation experiments using the anti-Regnase-1 antibody revealed that the interaction between Regnase-1 and -3 is likely to occur through the PIN domain of Regnase-1 (**Figure 4m**, reported below). This is consistent with previous studies showing that the PIN domain is important for the oligomerization of Regnase-1 (Yokogawa et al., 2016), and represents a novel finding of our study. This is now described also in the text (**page 8**). Finally, we found that the physical interaction between Regnase-1 and -3 did not require RNA binding, since it was observed also upon RNase treatment (**Supplementary Figure 7e**, reported below).

[Figures removed by editorial staff per authors' request]

The third concern is related to the evaluation of Regnase-1 degranulation in Figure 5g, where there appears to be some variability. Including Regnase-3 and conducting mutual evaluations could enhance the reliability of the results, possibly addressing this variability.

We apologize for our poor choice of data representation, that may have created confusion. First, we changed the representation of our data to a before-after histogram (**Figure for Reviewer** below), showing a very consistent and reproducible reduction in cell degranulation upon reduced Regnase-1 expression across different experiments and experimental systems (siRNAs and CRISPR-Cas9 RNPs).

[Figure removed by editorial staff per authors' request]

However, we previously attributed this effect of Regnase-1 on cell degranulation to a secondary consequence of the impact of Regnase-1 on cell proliferation/ survival (discussed on **Page 8** of the previous version of the manuscript). Indeed, we found that depletion or deletion of Regnase-3 did not impact cell degranulation.

[Figure removed by editorial staff per authors' request]

To confirm whether the small effect on cell degranulation observed upon deletion of Regnase-1 was indeed linked to the impaired viability of the cells, we repeated the degranulation experiments using a FACS-based, rather than colorimetric, system. Using annexin V staining (Leoni et al., 2017) we found that indeed there was no significant difference in cell degranulation, when gating only on alive cells, confirming that the previously observed small differences, albeit very reproducible, were indeed due the reduced survival of mast cells in absence of Regnase-1. We thank the Reviewer for pointing this out. To avoid confusion, we now removed panel 5g from the figure.

[Figure removed by editorial staff per authors' request]

Minor comments:

Figure 6e-g also seems to vary, and the effects on cell death and cell proliferation tend to be somewhat milder.

We apologize for our poor choice of data representation. We now changed the representation of our data to before-after histograms, showing a very consistent and reproducible effect of Regnase-1 deletion on cell survival and proliferation.

[Figure removed by editorial staff per authors' request]

Certainly, their IF images (Figure S2 and Figure S5) suggests that Regnase-1 and Regnase-3 have no or only a weak interaction.

We fully agree with the Reviewer. In fact, this aspect is specifically discussed in the manuscript **(Page 11)**.

Is the input to the immunoprecipitation from whole cell lysate or is it cleared with a low-speed (or high-speed) centrifugation?

We used pre-clearing with high-speed centrifugation. We now improved the description of this experiment in the Methods section.

Reviewer #1 (Significance (Required)):

The following aspects are important: Understanding the regulatory mechanisms mediated by RNA-binding proteins (RBPs) has gained significant attention in recent years. While it is inferred that RBPs control specific RNAs, many uncertainties remain about how they function in various RNA metabolism processes. The Regnase family is known to operate within the mechanism of RNA stability. While lower organisms have only one type, mammals have evolved to have four types. Understanding the implications of this family expansion is highly valuable, shedding light on how the RNA within our body's cells is regulated.

- Advance: In this study, a strength lies in the meticulous examination of the relationship between Regnase-1 and Regnase-3 by handling them similarly in the context of mast cells.

However, because of the multifaceted effects of Regnase-1 on cell proliferation, cell death, and the cell cycle, significant progress in understanding it has yet to be made. Going forward, the replication of immune responses in mast cells at the animal level holds the potential to further deepen our comprehension of RBPs through the Regnase family.

This study complements two previous investigations on Regnase-3 (PMID: 34215755, 31126966) while specifically focusing on mast cells. The findings align with the prevailing perspective that emphasizes the significance of Regnase-1 among the Regnase family.

- Audience: Basic research

- Immunology, Molecular Biology, Genetic Engineering

We thank the Reviewer for pointing out the importance and novelty of our study. Compared to the studies mentioned by the Reviewer, our work significantly advances the knowledge in the field in several important aspects: first, we focused on mast cells, a cell type for which very limited knowledge is available regarding post-transcriptional regulation. Second, we found a very clear 'division of labor' between Regnase-1 and -3, with Regnase-1 being required for homeostatic mast cell functions in the resting state, while both Regnase-1 and -3 cooperate in modulating inflammatory responses (see also our proposed model in the new **Supplementary Figure 9**). Third, we found that Regnase-1 is capable of direct protein-protein interaction with Regnase-3 in an RNA-independent manner, opening the possibility of additional post-transcriptional regulatory layers. Finally, we proved the importance of our findings in the regulation of human mast cells, pointing towards potential relevance for human disease.

Specifically, data mining of published RNA sequencing data revealed induction of *ZC3H12A* and *ZC3H12C* expression in stimulated human mast cells obtained from the skin (Gao et al., 2023). Similar results were obtained by RNA-seq of stimulated human peripheral blood-derived mast cells (Cildir et al., 2019) (**Supplementary Figure 1e**), highlighting the importance of Regnases also in the human system.

[Figure removed by editorial staff per authors' request].

Next, we used the human mast cell lines HMC 1.1 and 1.2 (Sundström et al., 2003), to assess the impact of Regnase-1 on TNF expression. We found that IVT mRNA transfection of human Regnase-1, but not a catalytically inactive version (D141N) significantly reduced TNF expression by human mast cells. Overall, our findings extend also to human cells, and are likely to be relevant in the context of human disease.

[Figure removed by editorial staff per authors' request]

Reviewer #2 (Evidence, reproducibility and clarity (Required)):

Summary:

Bataclan et al. studied the role of Regnase-1 and -3 in the mast cell (MC) function in vitro. Using siRNA, CRISPR-mediated depletion, and overexpression models in, mainly, IL-3-cultured bone-marrow-derived mast cells, they beautifully demonstrated that Regnase-1 and, to a lesser extent, Regnase-3 suppressed the expression of TNF in MCs in response to IgE-stimulation. Regnase-1 also had roles in MC survival and IgE-stimulation-induced degranulation. Regnase-3 seems to have mixed effects on MC functions as the protein's primary targets include both TNF and Regnase-1 mRNAs.

We thank the Reviewer for appreciating the high quality of our work.

Major comments:

1: The authors mainly used BMDCs generated by four weeks of cultivation of bone marrow cells with IL-3; at this point, MCs are considered "less mature". Could the authors reproduce the primary data (the role of Regnase-1 on TNF, cell survival, and degranulation) even if they used more mature MCs (e.g., extended cultivation with IL-3 (6-7 weeks))?

We apologize for the unclear description of this part of the Methods section, that we now improved. To clarify, we used BMDCs that were at least 4-weeks old. All our experiments were performed using mast cells between week 4 and week 8 of culture, with no difference detected across these different weeks.

2: TNF mRNA is not considered a primary target for Regnase-1 in other cell types, such as macrophages or fibroblasts. Although this reviewer does not doubt that the TNF expression level is controlled by Regnase-1 in MCs, more evidence is needed to conclude that TNF is a "primary target" for Regnases. They can determine the stability of "endogenous" TNF mRNA in Regnase-depleted cells, as in Figure 4d.

We thank the Reviewer for pointing this out. We performed additional experiments as recommended. We transfected mast cells with IVT mRNA-encoded Regnase-1 and -3 and we treated the cells with actinomycin D after IgE+antigen stimulation. We found that *Tnf* mRNA stability was significantly reduced in cells overexpressing Regnase-1, while the effect of Regnase-3 on *Tnf* stability was milder. These data are reported in the new **Figure 3j**, reported below, and are consistent with all our previous findings. We also performed a complementary experiment to Figure 4d, showing reduced *Zc3h12a* expression in cells transfected with IVT mRNA encoding Regnase-3 (**new Figure 4e**), further supporting our findings.

[Figure removed by editorial staff per authors' request]

Next, to assess whether the regulation of TNF expression by Regnase-1 in mast cells was conserved across species, we assessed the effect of IVT mRNA-transfected Regnase-1 in human mast cells. We used the human mast cell lines HMC 1.1 and 1.2 (Sundström et al., 2003) and found that IVT mRNA transfection of human Regnase-1, but not a catalytically inactive version (D141N) significantly reduced TNF expression by these cells (new **Supplementary Figure 6b-c**), further confirming the role of Regnase-1 in regulating TNF expression in mast cells.

[Figure removed by editorial staff per authors' request].

are currently part of another submitted manuscript, so they will not be included in the current manuscript.

Overall, our data are fully supportive of the regulation of *Tnf* expression by Regnase proteins, and we thank the Reviewer for allowing us to strengthen our conclusions.

[Figure removed by editorial staff per authors' request].

3: The role of Regnase-1 on the MC degranulation is less pronounced. Could the authors show the same result using another assay, such as a FACS-based assay excluding dead or dying cells?

We apologize for our poor choice of data representation, that may have created confusion. We first changed the representation of our data to a before-after histogram (**Figure for Reviewer**, reported below for convenience), showing a very consistent and reproducible reduction in the cell degranulation upon reduced Regnase-1 expression across different experiments and experimental systems (siRNAs and CRISPR-Cas9 RNPs).

[Figure removed by editorial staff per authors' request].

We previously attributed this effect of Regnase-1 on cell degranulation to a secondary consequence of the impact of Regnase-1 on cell proliferation/ survival (discussed on **page 8** of the previous version of the manuscript). Indeed, depletion or deletion of Regnase-3 did not impact cell degranulation (**Figure for Reviewer** below).

[Figure removed by editorial staff per authors' request].

To confirm whether the small effect on cell degranulation observed upon deletion of Regnase-1 was indeed linked to the impaired viability of the cells, we repeated the degranulation experiments using a FACS-based system, as recommended by the Reviewer. Using annexin V staining (Leoni et al., 2017) we found that indeed when gating only on alive cells, there was no significant difference in cell degranulation, confirming that the small differences previously observed, despite being very reproducible, were due the reduced survival of mast cells in absence of Regnase-1 (**Figure for Reviewer** below). We thank the Reviewer for pointing this out, which allowed us to clarify our findings. To avoid confusion, we now removed panel 5g from the figure.

[Figure removed by editorial staff per authors' request].

4: Have the authors had a chance to look into which protease(s) cleaved Regnase-1 in IgE-stimulated MCs?

Although we did not formally investigate which protease is responsible for Regnase-1 cleavage in IgE-stimulated mast cells, the paracaspase Malt-1 is known to cleave Regnase-1 in different systems, and it is also known to be activated downstream FcεRI signaling in IgE-stimulated mast cells (Klemm et al., 2006; Uehata et al., 2013). It is therefore highly likely that Malt1 is responsible for Regnase-1 cleavage in activated mast cells, as mentioned also in the text (**Page 4**).

Reviewer #2 (Significance (Required)):

General assessment:

This study investigated the role of Regnase-1 and -3 in the mast cell (MC) function in vitro. The strength of the study is that they used several methods to manipulate the expression levels of Regnases in the cells (siRNA, CRISPR, and Regnase-1 overexpression) and obtained consistent results. Although the study is well designed and the results are beautiful, the BMMCs the authors used are less mature MCs and do not represent the cells in the mammalian body well. Also, this study only focused on in vitro mouse-derived MCs, and human MCs or in vivo roles of MC Regnases were not studied.

We thank the Reviewer for pointing out the strengths and qualities of our study. We now provide additional data showing that our results are relevant also to human mast cells.

First, data mining of published RNA sequencing data revealed induction of *ZC3H12A* and *ZC3H12C* expression in stimulated human mast cells obtained from the skin (Gao et al., 2023). Similar results were obtained by RNA-seq of stimulated human peripheral blood-derived mast cells (Cildir et al., 2019) (**Supplementary Figure 1e**), highlighting the importance of Regnases also in the human system and in *ex-vivo* derived cells.

[Figure removed by editorial staff per authors' request].

Next, as discussed above, by using human mast cell lines we found that IVT mRNA transfection of human Regnase-1, but not a catalytically inactive version (D141N) significantly reduced TNF expression by human mast cells (new **Supplementary Figure 6b-c**). Overall, our findings fully extend to the human system, and are likely to be relevant in the context of human disease.

Advance:

The role of Regnase-1 in several cell types has already been shown, including the populations involved in type-2 immunity (Th2 and ILC2). However, this study might be the first to investigate

the role of Regnases in MCs. The roles of Regnases in MCs (control of mRNA stability through the RNase activity) presented are in line with the previous studies.

We thank the Reviewer for pointing out the importance and novelty of our study. Compared to previous studies, our work significantly advances the knowledge in the field in several important aspects: first, we focused on mast cells, a cell type for which very limited knowledge is available regarding post-transcriptional regulation. Second, we found a very clear 'division of labor' between Regnase-1 and -3, with Regnase-1 being required for homeostatic mast cell functions in the resting state, while both Regnase-1 and -3 cooperate in modulating inflammatory responses (see also our proposed model in the new **Supplementary Figure 9**). Third, we found that Regnase-1 is capable of direct protein-protein interaction with Regnase-3 in an RNA-independent manner, opening the possibility of additional post-transcriptional regulatory layers. Finally, we proved the importance of our findings in the regulation of human mast cells, pointing towards potential relevance for human disease.

Audience:

The primary audience of this study may be basic researchers studying type-2 immunity or MC biology. Because MCs are an essential cell population in several allergic disorders, some clinicians who care for allergic patients might also be interested in this study. However, main audiences may be relatively limited to specific fields like immunology and allergology.

We thank the Reviewer for pointing out that our study will be broadly relevant to the field of immunology.

This reviewer's main field of expertise is basic research in immunology and allergology. More specific keywords are MCs, ILC2, IgE, type-2 cytokines, and mouse models.

Reviewer #3 (Evidence, reproducibility and clarity (Required)):

Summary:

Bataclan et al. provide extensive in vitro work on Regnase1/3 function in mast cells. They make use of in vitro differentiated mast cells from bone marrow (BMMC) and in vitro cultures from primary peritoneal mast cells from mice. They show robustly Reg1/3 upregulation by transcript and protein upon activation stimulation in these cultured mast cells. Knockdown and CRISPR editing of Reg1/3 show enhanced inflammatory signaling. TNF is identified as a direct target of Reg1/3 nuclease activity. Reg3 is implicated as a negative regulator of Reg1. Overall, this work highlights Reg1/3 as novel players in mast cell biology that control inflammatory signaling including TNF.

We thank the Reviewer for pointing out the robustness of our data and the novelty of our conclusions.

Major comments:

- The result section requires more consistent & sufficient information on the assays (in vitro, ex vivo, species, etc.) particularly for the meta-analysis in the beginning. Relatedly, can the authors assume cross-species conservation of Reg1/3 in mast cells across mammals to make general conclusions on the functions or should this be limited mostly to mouse observations? It would be helpful to mention the species studied in the abstract.*

We apologize for the lack of clarity. We improved the description of the meta-analysis in the Results and Methods sections. We now mention the species studied in the abstract and added a paragraph to the Discussion highlighting how, given the extremely high species conservation between human and mouse Regnase-1 and -3 proteins (82.5% sequence identity for Regnase-1, 92% sequence identity for Regnase-3 between human and mouse), our results are likely to apply more broadly also to humans (**page 11-12**).

Indeed, data mining of published RNA sequencing data revealed induction of *ZC3H12A* and *ZC3H12C* expression in stimulated human mast cells obtained from the skin (Gao et al., 2023). Similar results were obtained by RNA-seq of stimulated human peripheral blood-derived mast cells (Cildir et al., 2019) (**Supplementary Figure 1e**, reported below), highlighting the importance of Regnases also in the human system and in *ex-vivo* derived cells.

[Figure removed by editorial staff per authors' request].

Concordant with this observation, we found that IVT mRNA transfection of human Regnase-1, but not a catalytically inactive version (D141N) significantly reduced TNF expression by human mast cells (new **Supplementary Figure 6b-c**). Overall, our findings fully extend to the human system, and are likely to be relevant in the context of human disease.

[Figure removed by editorial staff per authors' request].

• *Reg1 protein levels appear to be more stable than Reg3 protein levels (mirroring the transcript) Fig1a+c. Is the half-life of Reg3 shorter than Reg1? What is functional impact of Reg1 regulation at the transcriptional level considering the protein half-lives?*

The overall protein output of Regnase-1 and -3 is the result of transcriptional output, regulated at different extents by inflammatory signals (**Figure 1e**), the stability of the mRNA transcripts, the extent of translational output, and the stability of the proteins, also considering that at least Regnase-1 undergoes stimulation-dependent proteolytic cleavage (**Figure 1c-d**). Given the complexity of the regulation of these proteins, we have no way to disentangle the exact functional contribution of each of these regulatory steps to mast cell physiology. However, to clarify the novelty and impact of our findings, we now propose a model that distinguishes the resting and activated states of mouse and human mast cells. In the resting state, Regnase-3 is lowly expressed, while Regnase-1 expression is required to maintain basic mast cell homeostatic proliferation and survival. Upon activation, both proteins are strongly and dynamically induced, and they contribute to restrain excessive inflammatory responses (most notably TNF production). Regnase-3 induction also contributes to limit Regnase-1 expression, potentially favoring the re-establishment of a post-activation, quiescent state. This model is now included in **Supplementary Figure 9** and discussed in the text (**page 12**).

[Figure removed by editorial staff per authors' request].

• *MFI comparisons can only be used on unimodal populations (not bimodal i.e. Fig3b). Also, contour flow plots should show outliers.*

We thank the Reviewer for pointing this out. We substituted the MFI comparisons in **Figure 3b** and **3f** with cell percentages. We also show the outliers in the contour plots in **Figure 5b**.

Is TNF functionally secreted from WT vs KD/KO mast cells under these in vitro conditions? This is not shown and seems quite an important aspect given the focus on TNF regulation

(other secreted factors could also be considered). Degranulation appears to be lower in Fig5g. Does Reg1 overexpression reduce TNF expression and secretion? We performed an ELISA measurement for TNF using mast cells overexpressing Regnase-1 or Regnase-3, showing reduced TNF production that was more prominent for Regnase-1, fully consistent with our intracellular staining results (Figure 3h). The ELISA results are shown in the new Supplementary Figure 4c.

[Figure removed by editorial staff per authors' request].

- Catalytic variants of Reg1/3 appear to be only tested in WT cells. Wouldn't it be worthwhile to test these in KO cells? The absolute protein levels of ectopic and endogenous are not entirely clear and only shown for WT Reg3 in Fig4b.

The re-expression of Regnase proteins in knock-out mast cells is technically very challenging. First, knock-out populations must be generated by CRISPR-Cas9, leading unavoidably to a mixed population (we have no way to select effectively deleted cells). This mixed population must then be reconstituted transiently by IVT mRNA transfection, which also does not have an efficiency of 100%, leading to a very mixed population, with a lot of potentially confounding effects. Moreover, the deletion of Regnase-1 is sufficient to impair mast cell physiology quite dramatically, and IVT-mRNA expression of Regnase proteins is only stable for 1-2 days, hence the decision to use it only for very short-term experiments (cytokine expression).

To clarify what are the levels of expression of ectopic and endogenously expressed Regnase proteins in IVT mRNA experiments we performed a western blot analysis to compare mast cells transfected with IVT control mRNA (ZsGreen) to mast cells transfected with IVT mRNA encoding Regnase-3 and Regnase-1, wild-type or catalytically inactive. Cells were stimulated with IgE and antigen complexes for 4 h to induce endogenous Regnase protein expression. We found that the expression of

the IVT mRNA-derived proteins was much higher than the endogenous levels, that remained close to undetectable in these conditions therefore limiting the impact of the endogenously expressed proteins.

[Figure removed by editorial staff per authors' request]

• *Fig2b shows upregulation of Reg1 transcript is not enhanced upon Reg3 knockdown, but it is enhanced in Fig4a? How are these assays different?*

We apologize for the confusion. Knockdown or knockout of Regnase-3 led to similarly increased Regnase-1 protein expression in cells (Figure 2c and 3d). However, at the mRNA transcript level the effect of Regnase-3 deletion on Regnase-1 mRNA was more variable, especially in unstimulated cells, likely due to the low levels of expression. The difference between Figure 2b and Figure 4a was that in Figure 2b we showed the unstimulated cells, just as a control for the siRNAs, while in Figure 4a we showed the IgE-stimulated cells. To improve clarity, we now repeated some of these experiments and included both the resting and stimulated conditions for the siRNAs in Figure 2b. Our results show that indeed the *Zc3h12a* transcript is increased upon knock-down of Regnase-3. Such increase is statistically significant in stimulated cells, and more variable in resting conditions.

[Figure removed by editorial staff per authors' request]

• *The authors implicate Reg3 to regulate Reg1 by transcript. What is the role of the protein interaction between Reg1 and 3? Why is the protein interaction not seen in the IF experiments? Would other cell systems (maybe mouse) be more suitable.*

The physical interaction between Regnase-1 and -3 was identified in mouse mast cells and confirmed in human HEK293 cells. To further confirm and extend our findings we performed additional control immunoprecipitations using an anti-FLAG-tag antibody to immunoprecipitate FLAG-Regnase-1 or FLAG-Regnase-3, followed by detection with anti-Regnase-3 or anti-Regnase-1 antibody, respectively. We found that Regnase-3 co-immunoprecipitated with FLAG-Regnase-1 when using the anti-FLAG antibody (new **Supplementary Figure 7d**, reported below). Vice versa, Regnase-1 was not detectable upon immunoprecipitation of FLAG-Regnase-3 when using the anti-FLAG antibody, fully confirming our previous mass spectrometry data. We conclude that while Regnase-1 interacts with fewer proteins, including itself (Yokogawa et al., 2016) and Regnase-3, the protein-protein interactions of Regnase-3 are more promiscuous. Of these, Regnase-1 appears not be the predominant interactor, but rather proteins involved in translation, regulation of ribosomes and 14-3-3 proteins, consistent also with Regnase-1 and -3 being primarily located in different subcellular compartments, as revealed by our immunofluorescence experiments and discussed also in the text (**page 7 and 11**).

[Figure removed by editorial staff per authors' request]

Finally, to gain insights into the region(s) involved in protein-protein interactions between Regnase-1 and -3, we generated a panel of Regnase-3 C-terminal truncation mutants (schematic representation in **Figure 4k**). We found that all truncations were expressed at comparable levels in HEK293 cells (**Figure 4l**, reported below and new **Supplementary Figure 7e**). Co-immunoprecipitation experiments using the anti-Regnase-1 antibody revealed that the interaction between Regnase-1 and -3 is likely to occur through the PIN domain of Regnase-1 (**Figure 4m**, reported below).

[Figure removed by editorial staff per authors' request]

Our results are also consistent with previous studies showing that the PIN domain is important for the oligomerization of Regnase-1 (Yokogawa et al., 2016). This is now described also in the text (**page 8**).

• Cell death increases after Reg1/3 knockdowns; can this be rescued with Reg1/3 reinstation? Is the protein level of Reg1 important for this; can this be dosed? Are the assays for cell death/proliferation done in comparison to unstimulated resting cells, cultured conditions and IgE stimulated mast cells? Minor point: A single assay of cell death is sufficient in the main figures; same for proliferation assays.

As mentioned above, the re-expression of Regnase proteins in knock-out mast cells is technically very challenging. Especially the dying Regnase-1 knock-out cells are not amenable of being further transfected. Moreover, protein expression from IVT-delivered mRNA is stable only for a short period of time (hence the reason of performing only short-term, cytokine expression experiments). The cell death and proliferation assays were done in the absence of stimulation, we now clarified this point in the Methods section (**page 16**).

• It is not entirely clear how the RNA-seq data with CRISPR KO in Fig6 is different than the nanostring data with Knockdowns shown earlier in Fig2? Were unstimulated cells not used as a control before? A floxed mouse model for Reg1 is introduced at the end, but would have been useful for many assays.

The Nanostring panel was used only to compare the effects of Regnase perturbations upon stimulation with IgE and antigen complexes. The reason for this choice is that the Nanostring panel contains mostly probes for inflammatory transcripts that can be detected only upon stimulation, and very few genes for proliferation and survival, that were therefore analyzed by RNA-seq. We apologize for the confusion. We clarified this aspect in the text (**Page 5 and 8**). As for the floxed mouse, we fully agree with the Reviewer, but we only obtained cells from this mouse in recent months, so we decided to use them primarily as an important, additional confirmation of all our previous findings.

OPTIONAL: are in vivo experiments of interest to confirm the findings? The conclusions that Reg1/3 regulates physiological mast cell responses might be reached otherwise.

We attempted some *in vivo* studies using adoptive transfer and the passive cutaneous anaphylaxis system (Leoni et al., 2023). However, due to the complex effect of Regnase proteins on the mast cell phenotype and the cross-regulation between Regnase-3 and Regnase-1, our results showed a lot of variability, and no definitive conclusion could be drawn in this system. Further studies would require the generation of novel mouse models, that are beyond the current capabilities of our lab.

Minor comments:

- *Some of the Knockdown and CRISPR validation main figures are redundant and might be better suited for the supplement.*

Thank you, we think it is important to show these controls in the different conditions.

- *The MW in the western blot requires lines to indicate the exact location of the ladder.*

We added the exact location of the ladder in the western blots.

- *Fig2 title refers as loss of Reg1/3 using siRNA. The term is typically used in the context of genetic ablation and not for knockdowns.*

We modified the title of Figure 2 to “Depletion”, thank you.

- *Gene locus figures of Reg1/3 are confusing why are only E3 and E6 shown and not all Exons (Fig2a et al)?*

This was done just for simplicity, since several exons are separated by large introns, and they are not relevant to show the locations of the siRNAs or gRNAs, we clarified this in the Figure Legends.

- *Are biological replicates or technical replicates used in Fig1?*

All replicates shown in all figures (including Figure 1) are exclusively biological replicates. We clarified this in the Methods section.

- *Is a student's t test the appropriate statistical analysis for most of the analyses?*

A student's t test is appropriate for most of the analyses shown. We indicated in the Figure Legends whenever a different statistical test (for example ANOVA) was used.

- *An explanation why nanostring and RNA-seq are both used would be helpful.*

We improved the description and rationale of this choice, as detailed above (**page 5 and 8**).

Reviewer #3 (Significance (Required)):

Bataclan et al investigate Reg1/3 function in mast cells, that are of interest to elucidate the regulation of mast cell effector functions. Reg1/3 were identified as novel regulators of murine mast cells. Independent and complementary use of Knockdowns and CRISPR deletions provide robust data. Also, the use of two independent mast cell populations and different stimuli is of interest, albeit they are not used in every assay. Altogether, the in vitro data are robust and rigorous. On the flipside, in vivo data is not provided. The translational impact would be higher if findings are tested in in vivo models including preclinical applications (as discussed in text). Mouse studies are often indicative of mechanisms in higher mammals, but it would help to lay this out more clearly. Therefore, the connection to humans is less clear. The type of work presented here would be best described as basic research. This review has been assessed with the following expertise: mouse/human immunology, mouse models, immune cell signaling, immune effector functions, flow cytometry.

We thank the Reviewer for pointing out that our data are robust and rigorous and for highlighting the novelty of our study. As discussed above, we now extended our findings to the human system, pointing towards potential relevance in the context of human disease.

3. References

- Cildir, G., Toubia, J., Yip, K.H., Zhou, M., Pant, H., Hissaria, P., Zhang, J., Hong, W., Robinson, N., Grimbaldeston, M.A., *et al.* (2019). Genome-wide Analyses of Chromatin State in Human Mast Cells Reveal Molecular Drivers and Mediators of Allergic and Inflammatory Diseases. *Immunity* 51, 949-965.e946.
- Gao, J., Li, Y., Guan, X., Mohammed, Z., Gomez, G., Hui, Y., Zhao, D., Oskeritzian, C.A., and Huang, H. (2023). IL-33 priming and antigenic stimulation synergistically promote the transcription of proinflammatory cytokine and chemokine genes in human skin mast cells. *BMC genomics* 24, 592.
- Klemm, S., Gutermuth, J., Hültner, L., Sparwasser, T., Behrendt, H., Peschel, C., Mak, T.W., Jakob, T., and Ruland, J. (2006). The Bcl10-Malt1 complex segregates Fc epsilon RI-mediated nuclear factor kappa B activation and cytokine production from mast cell degranulation. *J Exp Med* 203, 337-347.
- Leoni, C., Bataclan, M., Ito-Kureha, T., Heissmeyer, V., and Monticelli, S. (2023). The mRNA methyltransferase Mettl3 modulates cytokine mRNA stability and limits functional responses in mast cells. *Nat Commun* 14, 3862.
- Leoni, C., Montagner, S., Rinaldi, A., Bertoni, F., Polletti, S., Balestrieri, C., and Monticelli, S. (2017). Dnmt3a restrains mast cell inflammatory responses. *Proc Natl Acad Sci U S A* 114, E1490-e1499.
- Sundström, M., Vliagoftis, H., Karlberg, P., Butterfield, J.H., Nilsson, K., Metcalfe, D.D., and Nilsson, G. (2003). Functional and phenotypic studies of two variants of a human mast cell line with a distinct set of mutations in the c-kit proto-oncogene. *Immunology* 108, 89-97.
- Uehata, T., Iwasaki, H., Vandenbon, A., Matsushita, K., Hernandez-Cuellar, E., Kuniyoshi, K., Satoh, T., Mino, T., Suzuki, Y., Standley, D.M., *et al.* (2013). Malt1-induced cleavage of regnase-1 in CD4(+) helper T cells regulates immune activation. *Cell* 153, 1036-1049.
- von Gamm, M., Schaub, A., Jones, A.N., Wolf, C., Behrens, G., Lichti, J., Essig, K., Macht, A., Pircher, J., Ehrlich, A., *et al.* (2019). Immune homeostasis and regulation of the interferon pathway require myeloid-derived Regnase-3. *J Exp Med* 216, 1700-1723.
- Yokogawa, M., Tsushima, T., Noda, N.N., Kumeta, H., Enokizono, Y., Yamashita, K., Standley, D.M., Takeuchi, O., Akira, S., and Inagaki, F. (2016). Structural basis for the regulation of enzymatic activity of Regnase-1 by domain-domain interactions. *Sci Rep* 6, 22324.

May 13, 2024

RE: Life Science Alliance Manuscript #LSA-2024-02784-T

Dr. Silvia Monticelli
Institute for Research in Biomedicine
Molecular Immunology Lab
Via Vincenzo Vela 6
Bellinzona 6500
Switzerland

Dear Dr. Monticelli,

Thank you for submitting your revised manuscript entitled "Crosstalk between the RNA-binding proteins Regnase-1 and -3 shapes mast cell survival and cytokine expression". We would be happy to publish your paper in Life Science Alliance pending final revisions necessary to meet our formatting guidelines.

- please consider Reviewer 2's comments
- please be sure that the authorship listing and order is correct
- please upload all figure files as individual ones, including the supplementary figure files; all figure legends should only appear in the main manuscript file
- please upload your main manuscript text as an editable doc file
- please add a Running Title and a Summary Blurb/Alternate Abstract to our system
- please add a Category for your manuscript in our system
- please add the Twitter handle of your host institute/organization as well as your own or/and one of the authors in our system
- please add the Author Contributions section to our system as well
- please add your main, supplementary figure, and table legends to the main manuscript text after the references section
- please upload your Tables in editable .doc or excel format
- you may want to consider uploading Figure S9 as a Graphical Abstract, rather than as a figure

A. FINAL FILES:

B. MANUSCRIPT ORGANIZATION AND FORMATTING:

Sincerely,

Reviewer #1 (Comments to the Authors (Required)):

I am satisfied with the author's revision.

Reviewer #2 (Comments to the Authors (Required)):

The authors have tried to address the concerns raised in the review, and as a result, the manuscript was significantly improved. The authors added new information regarding the role of Regnases in human mast cells (MCs). Also, the new results strengthened the authors' original statement that Tnf was the primary target for Regnase-1 in MCs.

The authors found that Regnase-1-mediated control of MC degranulation shown in the original manuscript was due to the poor survival of the Regnase-1 deficient cells and was not the direct effect of the Regnase-1 deficiency. Although the authors have decided to remove the data, this reviewer believes the data are still informative for readers. Because "degranulation" is always an indispensable topic in a MC-related study, the readers of the manuscript must want to know if Regnases control MC degranulation response.

A minor point

In the new Figure 3j, should the label on the y-axis "Zc3h12a mRNA remaining" be "Tnf mRNA remaining"?

Reviewer #3 (Comments to the Authors (Required)):

The authors have addressed the majority of the critiques. Thank you.

May 22, 2024

RE: Life Science Alliance Manuscript #LSA-2024-02784-TR

Dr. Silvia Monticelli
Institute for Research in Biomedicine
Molecular Immunology Lab
Via Francesco Chiesa 5
Bellinzona 6500
Switzerland

Dear Dr. Monticelli,

Thank you for submitting your Research Article entitled "Crosstalk between Regnase-1 and -3 shapes mast cell survival and cytokine expression". It is a pleasure to let you know that your manuscript is now accepted for publication in Life Science Alliance. Congratulations on this interesting work.

DISTRIBUTION OF MATERIALS:

Again, congratulations on a very nice paper. I hope you found the review process to be constructive and are pleased with how the manuscript was handled editorially. We look forward to future exciting submissions from your lab.

Sincerely,
